Journal of Data-centric Machine Learning Research (2024)          Submitted 10/12; Revised 2/19; Published 2/19

# Rethinking Symbolic Regression Datasets and Benchmarks for Scientific Discovery

**Yoshitomo Matsubara**[*]                                                              YOSHITOM@UCI.EDU
*Amazon Alexa, USA*

**Naoya Chiba**[†]                                                                      CHIBA@NCHIBA.NET
*Tohoku University, Japan*

**Ryo Igarashi**                                                                RYO.IGARASHI@SINICX.COM
*OMRON SINIC X Corporation, Japan*

**Yoshitaka Ushiku**                                                        YOSHITAKA.USHIKU@SINICX.COM
*OMRON SINIC X Corporation, Japan*

**Reviewed on OpenReview:** *https: // openreview. net/ forum? id= qrUdrXsiXX*

**Editor:** Theodoros Rekatsinas

## Abstract

This paper revisits datasets and evaluation criteria for Symbolic Regression (SR), specifically focused on its potential for scientific discovery. Focused on a set of formulas used in the existing datasets based on Feynman Lectures on Physics, we recreate 120 datasets to discuss the performance of symbolic regression for scientific discovery (SRSD). For each of the 120 SRSD datasets, we carefully review the properties of the formula and its variables to design reasonably realistic sampling ranges of values so that our new SRSD datasets can be used for evaluating the potential of SRSD such as whether or not an SR method can (re)discover physical laws from such datasets. We also create another 120 datasets that contain dummy variables to examine whether SR methods can choose necessary variables only. Besides, we propose to use normalized edit distances (NED) between a predicted equation and the true equation trees for addressing a critical issue that existing SR metrics are either binary or errors between the target values and an SR model's predicted values for a given input. We conduct benchmark experiments on our new SRSD datasets using various representative SR methods. The experimental results show that we provide a more realistic performance evaluation, and our user study shows that the NED correlates with human judges significantly more than an existing SR metric. We publish repositories of our code [1] and 240 SRSD datasets.[2] [3] [4] [5] [6] [7]

**Keywords:** symbolic regression for scientific discovery, physics, datasets, benchmarks

---

[*]. This work was mainly done while this author was a research intern at OMRON SINIC X Corporation.

[†]. This work was mainly done while this author was a project researcher at OMRON SINIC X Corporation.

1. `https://github.com/omron-sinicx/srsd-benchmark`

2. `https://huggingface.co/datasets/yoshitomo-matsubara/srsd-feynman_easy`

3. `https://huggingface.co/datasets/yoshitomo-matsubara/srsd-feynman_medium`

4. `https://huggingface.co/datasets/yoshitomo-matsubara/srsd-feynman_hard`

5. `https://huggingface.co/datasets/yoshitomo-matsubara/srsd-feynman_easy_dummy`

6. `https://huggingface.co/datasets/yoshitomo-matsubara/srsd-feynman_medium_dummy`

7. `https://huggingface.co/datasets/yoshitomo-matsubara/srsd-feynman_hard_dummy`

## 1 Introduction

Recent advances in machine learning (ML), especially deep learning, have led to the proposal of many methods that can reproduce the given data and make appropriate inferences on new inputs. Such methods are, however, often black-box, which makes it difficult for humans to understand how they made predictions for given inputs. This property will be more critical especially when non-ML experts apply ML to problems in their research domains such as physics and chemistry. Symbolic regression (SR) is the task of producing a mathematical expression (symbolic expression) that fits a given dataset. SR has been studied in the genetic programming (GP) community (Hoai et al., 2002; Keijzer, 2003; Koza and Poli, 2005; Johnson, 2009; Uy et al., 2011; Orzechowski et al., 2018), and deep learning-based SR methods have been attracting more attention from the machine learning community (Petersen et al., 2020; Landajuela et al., 2021; Biggio et al., 2021; Valipour et al., 2021; La Cava et al., 2021; Kamienny et al., 2022). Because of its interpretability, various scientific communities apply SR to advance research in their scientific fields *e.g.*, Physics (Wu and Tegmark, 2019; Udrescu and Tegmark, 2020; Udrescu et al., 2020; Kim et al., 2020; Cranmer et al., 2020; Liu and Tegmark, 2021; Liu et al., 2021), Applied Mechanics (Huang et al., 2021), Climatology (Abdellaoui and Mehrkanoon, 2021), Materials (Sun et al., 2019; Wang et al., 2019; Weng et al., 2020; Loftis et al., 2020), and Chemistry (Batra et al., 2020). Given that SR has been studied in various communities, La Cava et al. (2021) propose a benchmark framework for symbolic regression methods. In the benchmark study, they combine the Feynman Symbolic Regression Database (FSRD) (Udrescu and Tegmark, 2020) and the ODE-Strogatz repository (Strogatz, 2018) and compare a number of SR methods, using a large-scale heterogeneous computing cluster.[8]

To discuss the potential of symbolic regression for scientific discovery (SRSD), there still remain some issues to be addressed: oversimplified datasets and lack of evaluation metric towards SRSD. For symbolic regression tasks, existing datasets consist of minimum necessary variables and values sampled from limited domains *e.g.*, in range of 1 to 5, and there are no large-scale datasets with reasonably realistic values that capture the properties of the formula and its variables. Thus, it is difficult to discuss the potential of symbolic regression for scientific discovery using such existing datasets. For instance, the FSRD consists of 120 formulas selected mostly from Feynman Lectures Series[9] (Feynman et al., 1963a,b,c) and are core benchmark datasets used in SRBench (La Cava et al., 2021). While the formulas indicate physical laws, variables and constants in each dataset have no physical meanings since the datasets are not designed to discover the physical laws from the observed data in the real world. (See Section 3.1.)

Moreover, there is a lack of appropriate metrics to evaluate these methods for SRSD. An intuitive approach would be to measure the prediction error or correlation between the predicted values and the target values in the test data, as in standard regression problems. However, low prediction errors could be achieved even by complex models that differ from the original law. In addition, SRBench (La Cava et al., 2021) presents the percentage of agreement between the target and the estimated equations as solution rate. But in such cases,

---

8. Hosts with 24-28 core Intel(R) Xeon(R) CPU E5-2690 v4 2.60GHz processors and 250GB RAM.
9. Udrescu and Tegmark (2020) extract 20 of the 120 equations as "bonus" from other seminal books (Goldstein et al., 2002; Jackson, 1999; Weinberg, 1972; Schwartz, 2014).

both 1) equations that do not match at all and 2) that differ by only one term[10] are equally treated as incorrect. As a result, it is considered as a coarse-resolution evaluation method for accuracy in SRSD, which still needs more discussion towards real-world applications. A key feature of SR is its interpretability, and some studies (Udrescu et al., 2020; La Cava et al., 2021) use complexity of the predicted expression as an evaluation metric (the simpler the better). However, it is based on a big assumption that a simpler expression may be more likely to be a hidden law in the data (*e.g.*, physical law), which may not be true for SRSD. Therefore, there are no single evaluation metrics proposed to take into account both the interpretability and how close to the true expression the estimated expression is.

To address these issues, we propose new SRSD datasets, introduce a new evaluation method, and conduct benchmark experiments using various representative SR baseline methods. We carefully review and design annotation policies for the new datasets, considering the properties of the physics formulas. Besides, given that a formula can be represented as a tree structure, we introduce a normalized edit distance on the tree structure to allow quantitative evaluation of predicted formulas that do not perfectly match the true formulas. Using the proposed SRSD datasets and evaluation metric, we perform benchmark experiments with a set of SR baselines and find that there is still significant room for improvements in SRSD. Besides the datasets (Appendix A), we publish our code repository for reproducibility.

## 2 Related Studies

In this section, we briefly introduce related studies focused on 1) symbolic regression for scientific discovery and 2) symbolic regression dataset and evaluation.

### 2.1 SRSD: Symbolic Regression for Scientific Discovery

A pioneer study on symbolic regression for scientific discovery is conducted by Schmidt and Lipson (2009), who propose a data-driven scientific discovery method. They collect data from standard experimental systems like those used in undergrad physics education: an air-track oscillator and a double pendulum. Their proposed algorithm detects different types of laws from the data such as position manifolds, energy laws, and equations of motion and sum of forces laws.

Following the study, data-driven scientific discovery has been attracting attention from research communities and been applied to various domains such as Physics (Wu and Tegmark, 2019; Udrescu and Tegmark, 2020; Udrescu et al., 2020; Kim et al., 2020; Cranmer et al., 2020; Liu and Tegmark, 2021; Liu et al., 2021), Applied Mechanics (Huang et al., 2021), Climatology (Abdellaoui and Mehrkanoon, 2021), Materials (Sun et al., 2019; Wang et al., 2019; Weng et al., 2020; Loftis et al., 2020), and Chemistry (Batra et al., 2020). These studies leverage symbolic regression in different fields. While general symbolic regression tasks use synthetic datasets with limited sampling domains for benchmarks, many of the SRSD studies collect data from the real world and discuss how we could leverage symbolic regression toward scientific discovery.

While SRSD tasks share the same input-output interface with general symbolic regression (SR) tasks (*i.e.*, input: dataset, output: symbolic expression), we differentiate SRSD tasks in

---

10. If those differ by a constant or scalar, SRBench treats the estimated equation as correct for solution rate.

this study from general SR tasks by whether or not the datasets including true symbolic expressions are created with reasonably realistic assumptions for scientific discovery such as meaning of true symbolic expressions (whether or not they have physical meanings) and sampling domains for input variables.

## 2.2 Dataset and Evaluation

For symbolic regression methods, there exist several benchmark datasets and empirical studies. FSRD (Udrescu and Tegmark, 2020) is one of the largest symbolic regression datasets, which consists of 100 physics-inspired equations based on Feynman Lectures on Physics (Feynman et al., 1963a,b,c) and 20 equations from other seminal books (Goldstein et al., 2002; Jackson, 1999; Weinberg, 1972; Schwartz, 2014). By randomly sampling from small ranges of value, they generate the corresponding tabular datasets for the 120 equations. The ODE-Strogatz repository (La Cava et al., 2016) contains two-state dynamic models of first-order, ordinary differential equations sourced from (Strogatz, 2018). Inspired by Hoai et al. (2002); Keijzer (2003); Johnson (2009), Uy et al. (2011) suggest 10 different real-valued symbolic regression problems (functions) and create the corresponding dataset (*a.k.a.* Nguyen dataset). The suggested functions consist of either 1 or 2 variables *e.g.*, $f(x) = x^6 + x^5 + x^4 + x^3 + x^2 + x$ and $f(x, y) = \sin(x) + \sin(y^2)$. They generate each dataset by randomly sampling 20 - 100 data points. La Cava et al. (2021) design a symbolic regression benchmark, named SRBench, and conduct a comprehensive benchmark experiment, using existing symbolic regression datasets such as FSRD and ODE-Strogatz repository (La Cava et al., 2016). In SRBench, SR methods are assessed by 1) an error metric based on squared error between target and estimated values, and 2) solution rate that shows a percentage of the estimated symbolic regression models that match the true models based on sympy (Meurer et al., 2017). However, these datasets and evaluations are not necessarily designed to discuss the potential of SRSD. In Sections 3.1 and 4.1, we further describe potential issues in prior studies.

## 3 Datasets

We summarize issues we found in the existing symbolic regression datasets, and then propose new datasets to address them towards symbolic regression for scientific discovery (SRSD).

## 3.1 Issues in Existing Datasets

As introduced in Section 2.2, there are many symbolic regression datasets. However, we consider that novel datasets are required to discuss SRSD for the following reasons:

1. **No physical meaning:** Many of the existing symbolic regression datasets (Hoai et al., 2002; Keijzer, 2003; Johnson, 2009; Uy et al., 2011; Trujillo et al., 2016; Jin et al., 2019) are not necessarily physics-inspired, but instead randomly generated *e.g.*, $f(x) = \log(x)$, $f(x, y) = xy + \sin((x - 1)(y - 1))$. To discuss the potential of SRSD, we need to further elaborate datasets and evaluation metrics, considering how we would leverage symbolic regression in practice.

2. **Oversimplified sampling process:** While some of the datasets are physics-inspired such as FSRD (Udrescu and Tegmark, 2020) and ODE-Strogatz repository (La Cava

et al., 2016), their sampling strategies are very simplified. Specifically, the strategies do not distinguish between constants and variables *e.g.*, speed of light[11] is treated as a variable and randomly sampled in range of 1 to 5. Besides, most of the sampling domains are far from values we could observe in the real world *e.g*, II.4.23 in Table 8 (the vacuum permittivity values are sampled from range of 1 to 5). When sampled ranges of the distributions are narrow, we cannot distinguish Lorentz transformation from Galilean transformation *e.g.* I.15.10 and I.16.6 in Table 10, I.48.2 in Table 12, I.15.3t, I.15.3x, and I.34.14 in Table 14, or the black body radiation can be misestimated to Stephan-Boltzmann law or the Wien displacement law *e.g.* I.41.16 in Table 15.

3. **Duplicate SR problems:** Due to the two issues above, many of the equations in existing datasets turn out to be duplicate. *e.g.*, as shown in Table 2, $F = \mu N_n$ (I.12.1) and $F = q_2 E$ (I.12.5) in the *original* FSRD are considered identical since both the equations are multiplicative and consists of two variables, and their sampling domains (Distributions in Table 2) are exactly the same. For instance, approximately 25% of the symbolic regression problems in the *original* FSRD have 1 - 5 duplicates in that regard.

4. **Incorrect/Inappropriate formulas:** FSRD (Udrescu and Tegmark, 2020) treat every variables as float whereas they should be integer to be physically meaningful. For example, the difference in number of phases in Bragg's law should be integer but sampled as real number (I.30.5 in Table 8). Furthermore, they do not even give special treatment of angle variables (I.18.12, I.18.16, and I.26.2 in Table 2). Physically some variables can be negative whereas the *original* FSRD only samples positive values (*e.g.* I.8.14 and I.11.19 in Table 10). We also avoid using *arcsin/arccos* in the equations since the use of *arcsin/arccos* in FSRD just to obtain angle variables is not experimentally meaningful (I.26.2 in Table 2, I.30.5 in Table 8, and B10 in Table 18). Equations using *arcsin* and *arccos* in the original annotation are I.26.2 (Snell's law), I.30.5 (Bragg's law), and B10 (Relativistic aberration). These are all describing physical phenomena related to two angles, and it is an unnatural deformation to describe only one of them with an inverse function. Additionally, inverse function use implicitly limits the range of angles, but there is no such limitation in the actual physical phenomena.

5. **Ignoring feature selection:** The existing SR datasets consist of samples using only necessary input variables to symbolically express the true models. *E.g.*, if the true model is $F = \mu N_n$ (I.12.1) in Table 2, an existing SR dataset would consist of three variables only (*i.e.*, a three-column tabular dataset): two input variables $\mu$ (coefficient of friction), $N_n$ (normal force), and the target variable $F$ (force of friction). Suppose we do not know the physics law $F = \mu N_n$. When we observe scenes of the system using some experimental tools, we may measure other input variables 1) ground contact area of the object $a$ and 2) velocity of the object $v$ in addition to $\mu$, $N_n$, and $F$. When we want to discover the physics law from the observed data points (a five-column tabular dataset), both $a$ and $v$ play the same role as dummy variables and should be excluded through feature selections. SR methods should be able to select the only necessary input variables or features ($\mu$, $N_n$) from the given data ($\mu$, $N_n$, $a$, $v$, and $w$), but we cannot discuss such robustness of SR methods, using the existing SR datasets.

---

11. We treat speed of light as a constant ($2.998 \times 10^8$ [m/s]).

Table 1: SR dataset comparisons with respect to issues summarized in Section 3.1. (✗: not addressed, ✓: addressed)

| Dataset | #problems | Issue 1 | Issue 2 | Issue 3 | Issue 4 | Issue 5 |
|---|---|---|---|---|---|---|
| (Hoai et al., 2002) | 4 | ✗ | ✗ | ✓ | ✓ | ✗ |
| (Keijzer, 2003) | 15 | ✗ | ✗ | ✗ | ✓ | ✗ |
| (Johnson, 2009) | 7 | ✗ | ✗ | ✓ | ✓ | ✗ |
| (Uy et al., 2011) | 10 | ✗ | ✗ | ✓ | ✓ | ✗ |
| (Trujillo et al., 2016) | 9 | ✗ | ✗ | ✓ | ✓ | ✗ |
| (La Cava et al., 2016) | 10 | ✓ | ✗ | ✓ | ✓ | ✗ |
| (Jin et al., 2019) | 6 | ✗ | ✗ | ✓ | ✓ | ✗ |
| (Udrescu and Tegmark, 2020) | 120 | ✓ | ✗ | ✗ | ✗ | ✗ |
| **Ours: SRSD-Feynman** | 240 | ✓ | ✓ | ✓ | ✓ | ✓ |

## 3.2 Proposed SRSD Datasets

We address the issues in existing datasets above by proposing new SRSD datasets based on the equations used in the FSRD (Udrescu and Tegmark, 2020). Section 3.1 and Table 1 summarize the differences between the FSRD and our SRSD datasets. Our annotation policy is carefully designed to simulate typical physics experiments so that the SRSD datasets can engage studies on symbolic regression for scientific discovery in the research community.

### 3.2.1 ANNOTATION POLICY

We thoroughly revised the sampling range for each variable from the annotations in the FSRD (Udrescu and Tegmark, 2020). First, we reviewed the properties of each variable and treated physical constants (*e.g.*, speed of light, gravitational constant) as constants while such constants are treated as variables in the original FSRD datasets (**Issues 1, 4**). As shown in Table 2, it also makes I.12.1 and I.12.5 two separate problems in SRSD datasets while these two problems in the original FSRD are duplicates because both the problems share the identical symbolic expressions and sampling ranges. Next, we defined sampling ranges in SI units to correspond to each typical physics experiment to confirm the physical phenomenon for each equation (**Issues 2, 3, 4**). We referenced (Feynman et al., 1963a,b,c; National Astronomical Observatory of Japan, 2022) to understand the context in which each formula appeared in our datasets. Taking mass as an example, it can be the mass of the Earth, an atom, or something else, depending on the context of each formula. In cases where a specific experiment is difficult to be assumed, ranges were set within which the corresponding physical phenomenon can be seen. Generally, the ranges are set to be sampled on log scales within their orders as $10^2$ in order to take both large and small changes in value as the order changes. Variables such as angles, for which a linear distribution is expected are set to be sampled uniformly. In addition, variables that take a specific sign were set to be sampled within that range. Tables 2 and 8 – 18 show the detailed comparisons between the original FSRD and our proposed SRSD datasets. We also build another 120 SRSD datasets,

which contain dummy variables to discuss the robustness of SR methods against dummy variables (**Issue 5**), and there will be 240 proposed datasets in total.[12] See Section 3.3 for the detail of the datasets with dummy variables.

Table 2: Easy set of our proposed datasets (part 1). C: Constant, V: Variable, F: Float, I: Integer, P: Positive, N: Negative, NN: Non-Negative, $\mathcal{U}$: Uniform distribution, $\mathcal{U}_{\log}$: Log-Uniform distribution. Other 110 datasets are summarized in Tables 8 - 18.

| Eq. ID | Formula | | Symbols | Properties | | Distributions | |
|---|---|---|---|---|---|---|---|
| | | | | Original | Ours | Original | Ours |
| I.12.1 | $F = \mu N_{\mathrm{n}}$ | $F$ | Force of friction | V, F, P | V, F, P | N/A | N/A |
| | | $\mu$ | Coefficient of friction | V, F, P | V, F, P | $\mathcal{U}(1,5)$ | $\mathcal{U}_{\log}(10^{-2}, 10^0)$ |
| | | $N_{\mathrm{n}}$ | Normal force | V, F, P | V, F, P | $\mathcal{U}(1,5)$ | $\mathcal{U}_{\log}(10^{-2}, 10^0)$ |
| I.12.4 | $E = \frac{q_1}{4\pi\epsilon r^2}$ | $E$ | Magnitude of electric field | V, F, P | V, F | N/A | N/A |
| | | $q_1$ | Electric charge | V, F, P | V, F | $\mathcal{U}(1,5)$ | $\mathcal{U}_{\log}(10^{-1}, 10^1)$ |
| | | $r$ | Distance | V, F, P | V, F, P | $\mathcal{U}(1,5)$ | $\mathcal{U}_{\log}(10^{-1}, 10^1)$ |
| | | $\epsilon$ | Vacuum permittivity | V, F, P | C, F, P | $\mathcal{U}(1,5)$ | $8.854 \times 10^{-12}$ |
| I.12.5 | $F = q_2 E$ | $F$ | Force | V, F, P | V, F | N/A | N/A |
| | | $q_2$ | Electric charge | V, F, P | V, F | $\mathcal{U}(1,5)$ | $\mathcal{U}_{\log}(10^{-1}, 10^1)$ |
| | | $E$ | Electric field | V, F, P | V, F | $\mathcal{U}(1,5)$ | $\mathcal{U}_{\log}(10^{-1}, 10^1)$ |
| I.14.3 | $U = mgz$ | $U$ | Potential energy | V, F, P | V, F | N/A | N/A |
| | | $m$ | Mass | V, F, P | V, F, P | $\mathcal{U}(1,5)$ | $\mathcal{U}_{\log}(10^{-2}, 10^0)$ |
| | | $g$ | Gravitational acceleration | V, F, P | C, F, P | $\mathcal{U}(1,5)$ | $9.807 \times 10^0$ |
| | | $z$ | Height | V, F, P | V, F | $\mathcal{U}(1,5)$ | $\mathcal{U}_{\log}(10^{-2}, 10^0)$ |
| I.14.4 | $U = \frac{k_{\mathrm{spring}} x^2}{2}$ | $U$ | Elastic energy | V, F, P | V, F, P | N/A | N/A |
| | | $k_{\mathrm{spring}}$ | Spring constant | V, F, P | V, F, P | $\mathcal{U}(1,5)$ | $\mathcal{U}_{\log}(10^2, 10^4)$ |
| | | $x$ | Position | V, F, P | V, F | $\mathcal{U}(1,5)$ | $\mathcal{U}_{\log}(10^{-2}, 10^0)$ |
| I.18.12 | $\tau = rF\sin\theta$ | $\tau$ | Torque | V, F | V, F | N/A | N/A |
| | | $r$ | Distance | V, F, P | V, F, P | $\mathcal{U}(1,5)$ | $\mathcal{U}_{\log}(10^{-1}, 10^1)$ |
| | | $F$ | Force | V, F, P | V, F, P | $\mathcal{U}(1,5)$ | $\mathcal{U}_{\log}(10^{-1}, 10^1)$ |
| | | $\theta$ | Angle | V, F, NN | V, F, NN | $\mathcal{U}(0,5)$ | $\mathcal{U}(0,2\pi)$ |
| I.18.16 | $L = mrv\sin\theta$ | $L$ | Angular momentum | V, F | V, F | N/A | N/A |
| | | $m$ | Mass | V, F, P | V, F, P | $\mathcal{U}(1,5)$ | $\mathcal{U}_{\log}(10^{-1}, 10^1)$ |
| | | $r$ | Distance | V, F, P | V, F, P | $\mathcal{U}(1,5)$ | $\mathcal{U}_{\log}(10^{-1}, 10^1)$ |
| | | $v$ | Velocity | V, F, P | V, F, P | $\mathcal{U}(1,5)$ | $\mathcal{U}_{\log}(10^{-1}, 10^1)$ |
| | | $\theta$ | Angle | V, F, P | V, F, NN | $\mathcal{U}(1,5)$ | $\mathcal{U}(0,2\pi)$ |
| I.25.13 | $V = \frac{q}{C}$ | $V$ | Voltage | V, F, P | V, F | N/A | N/A |
| | | $q$ | Electric charge | V, F, P | V, F | $\mathcal{U}(1,5)$ | $\mathcal{U}_{\log}(10^{-5}, 10^{-3})$ |
| | | $C$ | Electrostatic Capacitance | V, F, P | V, F, P | $\mathcal{U}(1,5)$ | $\mathcal{U}_{\log}(10^{-5}, 10^{-3})$ |
| I.26.2 | $n = \frac{\sin\theta_1}{\sin\theta_2}$ | $n$ | Relative refractive index | V, F, NN | V, F, P | $\mathcal{U}(0,1)$ | N/A |
| | | $\theta_1$ | Refraction angle 1 | V, F | V, F, NN | N/A | $\mathcal{U}(0, \frac{\pi}{2})$ |
| | | $\theta_2$ | Refraction angle 2 | V, F, P | V, F, NN | $\mathcal{U}(1,5)$ | $\mathcal{U}(0, \frac{\pi}{2})$ |
| I.27.6 | $f = \frac{1}{\frac{1}{d_1} + \frac{n}{d_2}}$ | $f$ | Focal length | V, F, P | V, F, P | N/A | N/A |
| | | $d_1$ | Distance | V, F, P | V, F, P | $\mathcal{U}(1,5)$ | $\mathcal{U}_{\log}(10^{-3}, 10^{-1})$ |
| | | $n$ | Refractive index | V, F, P | V, F, P, | $\mathcal{U}(1,5)$ | $\mathcal{U}_{\log}(10^{-1}, 10^1)$ |
| | | $d_2$ | Distance | V, F, P | V, F, P | $\mathcal{U}(1,5)$ | $\mathcal{U}_{\log}(10^{-3}, 10^{-1})$ |

12. We provide scripts to generate the datasets as part of our code repository.

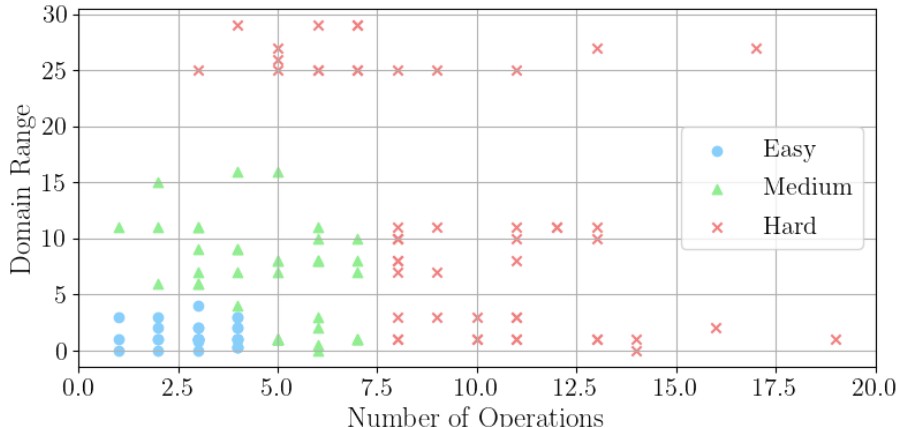

Figure 1: Distribution map of three subsets for our SRSD datasets with respect to our complexity metrics of SR problem. Data points at top right/bottom left indicate more/less complex problems.

### 3.2.2 COMPLEXITY-AWARE DATASET CATEGORIES

While the proposed datasets consist of 120 different problems, there will be non-trivial training cost required to train a symbolic regression model for all the problems individually (La Cava et al., 2021) *i.e.*, there will be 120 separate training sessions to assess the symbolic regression approach. To allow more flexibility in assessing symbolic regression models for scientific discovery, we define three clusters of the proposed datasets based on their complexity: *Easy*, *Medium*, and *Hard* sets, which consist of 30, 40, and 50 different problems respectively.

We define the complexity of a problem, using the number of operations to represent the true equation tree and range of the sampling domains. The former measures how many mathematical operations compose the true equation such as *add*, *mul*, *pow*, *exp*, and *log* operations (see Fig. 2). The latter considers magnitude of sampling distributions (*Distributions* column in Tables 2 and 8 – 18) and increases the complexity when sampling values from wide range of distributions. We define the domain range as follows:

$$f_{\text{range}}\left(\mathcal{S}\right) = \left| \log_{10} \left| \max_{s \in \mathcal{S}} s - \min_{s \in \mathcal{S}} s \right| \right|, \tag{1}$$

where $\mathcal{S}$ indicates a set of sampling domains (*distributions*) for a given symbolic regression problem. Using these two metrics, we define *Easy*, *Medium*, or *Hard* sets as illustrated in Fig. 1.

These clusters represent problem difficulties at high level. For instance, these subsets will help the research community to shortly tune and/or perform sanity-check new approaches on the *Easy* set (30 problems) instead of using the whole datasets (120 problems). Figure 1 shows the three different distribution maps of our proposed datasets.

### 3.3 Introducing Dummy Variables to Our SRSD Datasets

As pointed out in Section 3.1, existing SR datasets such as the FSRD consist of only necessary variables to express the predicted equation. However, there may be irrelevant features (input variables) in the observed samples for real-world applications, and then SR methods should detect and exclude such input variables from their predicted solutions (equations). The existing SR datasets are not suitable for benchmarking SR methods from the aspect, especially for SRSD problems, and thus we introduce dummy variables to our SRSD datasets:

1. Given an SRSD dataset (input variables + target variable), randomly choose $k_{\text{dummy}}$, the number of dummy variables to be introduced, from $\{1, 2, 3\}$.

2. For each of the $k_{\text{dummy}}$ dummy variables,

    2.1. randomly choose the index of the dummy variable (column index of the dummy variable in the resulting tabular dataset),

    2.2. randomly determine (with a probability of 50%) whether or not the dummy variable can be sampled from negative sampling range, and

    2.3. randomly choose $s$ from $\{-32, -31, -30, \ldots, 30, 31, 32\}$ and sample $N$ values from $\mathcal{U}_{\log}(10^{s-1}, 10^{s+1})$ for the dummy variable, where $N$ is the number of samples in the given SRSD dataset.

We apply the above procedure to each of the 120 SRSD-Feynman datasets independently. Thus, each of the 120 new SRSD datasets ("SRSD-Feynman + Dummy Variables" in Table 5) will have a different configuration: number of dummy variables, indices of the dummy variables, and their sampling ranges. Table 3 summarizes how many random dummy variables were introduced to which datasets (equations).

## 4 Benchmark

Besides the conventional metrics, we propose a new metric to discuss the performance of symbolic regression for scientific discovery in Section 4.1. Following the set of metrics, we design an evaluation framework of symbolic regression for scientific discovery, hoping that the proposed SRSD benchmark helps non-ML experts choose SR methods for their problems.

### 4.1 Metrics

In general, it would be difficult to define "accuracy" of symbolic regression models since we will compare its estimated equation to the ground truth equation and need criteria to determine whether or not it is "correct". La Cava et al. (2021) suggest a reasonable definition of symbolic solution, which is designed to capture symbolic regression models that differ from the true model by a constant or scalar.[13] They also use $R^2$ score (Eq. 2) and define as accuracy the percentage of symbolic regression problems that a model meets $R^2 > \tau$, where $\tau$ is a threshold $e.g.$, $\tau = 0.999$ in (La Cava et al., 2021):

$$R^2 = 1 - \frac{\sum_j^N \left(f_{\text{pred}}\left(X_{\text{test},j}\right) - f_{\text{true}}\left(X_{\text{test},j}\right)\right)^2}{\sum_k^N \left(f_{\text{true}}\left(X_{\text{test},k}\right) - \bar{y}\right)^2}, \tag{2}$$

---

13. Code in La Cava et al. (2021) ignores coefficient terms whose absolute values are less than $10^{-4}$.

Table 3: Equation IDs and numbers of dummy variables introduced to our SRSD datasets.

| Group | 1 dummy variable | 2 dummy variables | 3 dummy variables |
|-------|------------------|-------------------|-------------------|
| Easy | I.12.1, I.12.4, I.12.5, I.18.12, I.25.13, I.47.23 | I.14.3, I.18.16, I.43.16, II.3.24, II.8.31, II.10.9, II.13.17, II.15.5, II.27.18, III.7.38, III.12.43 | I.14.4, I.26.2, I.27.6, I.30.5, II.2.42, II.4.23, II.15.4, II.27.16, II.34.11, II.34.29b, II.38.3, II.38.14, III.15.27 |
| Medium | I.10.7, I.12.2, I.13.12, I.16.6, I.32.5, I.43.31, II.11.3, II.34.2, II.34.29a, III.14.14, III.15.14, B8 | I.11.19, I.12.11, I.13.4, I.15.10, I.18.4, I.24.6, I.34.8, I.38.12, I.39.11, I.43.43, I.48.2, II.6.11, II.21.32, II.34.2a, III.4.32, III.13.18, III.15.12, III.17.37 | I.8.14, I.29.4, I.34.10, I.34.27, I.39.10, II.8.7, II.37.1, III.8.54, III.19.51, B18 |
| Hard | I.15.3x, I.30.3, II.6.15a, II.11.17, II.11.28, II.13.23, II.13.34, II.24.17, B1, B6, B12, B16, B17 | I.6.20, I.6.20b, I.9.18, I.15.3t, I.29.16, I.34.14, I.39.22, I.44.4, II.11.20, II.11.27, II.35.18, III.9.52, III.10.19, III.21.20, B2, B3, B7, B9 | I.6.20a, I.32.17, I.37.4, I.40.1, I.41.16, I.50.26, II.6.15b, II.35.21, II.36.38, III.4.33, B4, B5, B10, B11, B13, B14, B15, B19, B20 |

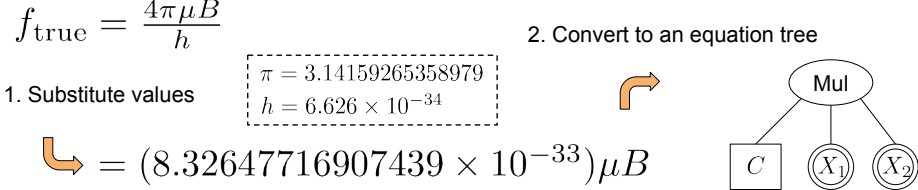

$$f_{\text{true}} = \frac{4\pi\mu B}{h}$$

1. Substitute values

$\pi = 3.14159265358979$
$h = 6.626 \times 10^{-34}$

2. Convert to an equation tree

$= (8.32647716907439 \times 10^{-33})\mu B$

Figure 2: Example of preprocessing a true equation (III.7.38 in Table 8) in evaluation session. When converting to an equation tree, we replace constant values and variables with specific symbols $e.g.$, $8.32647716907439 \times 10^{-33} \rightarrow C, \mu \rightarrow X_1, B \rightarrow X_2$.

where $N$ indicates the number of test samples ($i.e.$, the number of rows in the test dataset $X_{\text{test}}$), and $X_{\text{test},i}$ indicates the $i$-th test sample. $\bar{y}$ is a mean of target outputs produced by $f_{\text{true}}$. $f_{\text{pred}}$ and $f_{\text{true}}$ are a trained SR model and a true model, respectively. However, these two metrics are still binary (correct or not) or require a threshold and do not explain how $structurally$ $close$ to the true equation the estimated one is. While a key feature of symbolic regression is its interpretability, there are no single evaluation metrics to take into account both the interpretability and how close to the true expression the estimated expression is.

To offer more flexibility and assess estimated equations in such a way, we propose use of edit distance between estimated and ground truth equations, processing equations as trees. Although edit distance has been employed in different domains such as machine translation (Przybocki et al., 2006) (text-based edit distance), its primary use has been to study the search process for genetic programming approaches (O'Reilly, 1997; Burke et al., 2002; Nakai et al., 2013). Different from prior work, we propose a use of tree-based edit

distance as a new metric of solution quality for SRSD. For a pair of two trees, edit distance computes the minimum cost to transform one to another with a sequence of operations, each of which either 1) *inserts*, 2) *deletes*, or 3) *renames* a node. In this study, a node can be either a mathematical operation (*e.g.*, *add*, *exp* as symbols), a variable symbol, or a constant symbol. For the detail of the algorithm, we refer readers to (Zhang and Shasha, 1989).

As illustrated in Fig. 2, we preprocess equations by 1) substituting constant values *e.g.*, $\pi$ and Planck constant to the expression, and 2) converting the resulting expression to an equation tree that represents the preorder traversal of the equation with simplified symbols. It should be worth noting that before generating the equation tree, we simplify and convert equations to floating-point approximations[14] by sympy Meurer et al. (2017), a Python library for symbolic mathematics. It helps us consistently map a given equation to the unique equation tree and compute edit distance between the true and estimated equation trees since our evaluation interest is in simplified expressions of the estimated equations rather than how SR models produced the equations. For instance, "$x + x + x$", "$4 * x - x$", and "$x + 2 * x$" will be simplified by sympy to "$3 * x$" and considered identical.

For edit distance, we use a method proposed by Zhang and Shasha (1989). Given that the range of edit distance values depends on complexity of equations, we normalize the distance in range of 0 to 1 as

$$\bar{d}(f_{\text{pred}}, f_{\text{true}}) = \min\left(1, \frac{d\left(f_{\text{pred}}, f_{\text{true}}\right)}{|f_{\text{true}}|}\right), \tag{3}$$

where $f_{\text{pred}}$ and $f_{\text{true}}$ are estimated and true equation trees, respectively. $d(f_{\text{pred}}, f_{\text{true}})$ is an edit distance between $f_{\text{pred}}$ and $f_{\text{true}}$. $|f_{\text{true}}|$ indicates the number of the tree nodes that compose an equation $f_{\text{true}}$. We note that this metric is designed to capture similarity between estimated and true equations, thus coefficient values themselves (*e.g.*, value of $C$ in Fig. 2) should not be important.

## 4.2 Evaluation Framework

For each problem, we use the validation tabular dataset and choose the best trained SR model $f_{\text{pred}}^*$ from $\mathcal{F}$, a set of the trained models by a given method respect to Eq. (4)

$$f_{\text{pred}}^* = \arg\min_{f_{\text{pred}} \in \mathcal{F}} \frac{1}{n} \sum_{i=1}^{n} \left| \frac{f_{\text{pred}}(X_{\text{val},i}) - f_{\text{true}}(X_{\text{val},i})}{f_{\text{true}}(X_{\text{val},i})} \right|^2, \tag{4}$$

where $X_{\text{val},i}$ indicates the $i$-th row of the validation tabular dataset $X_{\text{val}}$.

In our SRSD-Feynman, we provide true equations as part of test datasets besides tabular data for benchmark purposes. In real-world applications, however, only observed samples (*e.g.*, tabular data) are available for training, validation, and test. Notice that similar to the solution rate proposed in (La Cava et al., 2021), normalized edit distance (NED) requires both the predicted and true equations. For this reason, we use the geometrical distance between predicted values against a validation tabular dataset to choose the best model obtained through hyperparameter tuning. Using the best model per method, we compute the normalized edit distance to assess the given method.

---

14. `https://github.com/omron-sinicx/srsd-benchmark/blob/main/eq_comparator.py`

## 5 Experiments

### 5.1 Baseline Methods

For baselines, we use the five best symbolic regression methods in SRBench (La Cava et al., 2021). Specifically, we choose gplearn (Koza and Poli, 2005), AFP (Schmidt and Lipson, 2011), AFP-FE (Schmidt and Lipson, 2009), AI Feynman (AIF) (Udrescu et al., 2020), and DSR (Petersen et al., 2020), referring to the rankings of solution rate for the FSRD datasets in their study. We note that La Cava et al. (2021) also benchmark symbolic regression methods for black-box problems,[15] whose true symbolic expressions are unknown, and other symbolic regression methods *e.g.*, Operon (Kommenda et al., 2020), SBP-GP (Virgolin et al., 2019), FEAT (La Cava et al., 2018), EPLEX (La Cava et al., 2019), and GP-GOMEA (Virgolin et al., 2021) outperform the five baseline methods we choose from their study, in terms of $R^2$-based accuracy. However, we find solution rate more aligned with edit distance, thus we choose the five best symbolic regression methods in terms of solution rate empirically shown for the FSRD datasets in SRBench (La Cava et al., 2021). We also use three symbolic regression methods proposed in recent studies: a Transformer-based symbolic regression method referred to as E2E in (Kamienny et al., 2022), unified deep symbolic regression (uDSR) (Landajuela et al., 2022), and a multi-population evolutionary algorithm named PySR (Cranmer, 2023). For details of the baseline models, we refer readers to the corresponding papers (Koza and Poli, 2005; Schmidt and Lipson, 2011, 2009; Udrescu et al., 2020; Petersen et al., 2020; Kamienny et al., 2022; Landajuela et al., 2022; Cranmer, 2023). While we are aware that the research community is interested in performance of closed API services powered by language models such as ChatGPT [16], it is not known what datasets are used to train the models behind the services. We do not consider such services in this study since the benchmark uses our SRSD datasets on popular physics laws.[17]

### 5.2 Runtime Constraints

The implementations of the baseline methods in Section 5.1 except E2E[18] do not use any GPUs. We run 1,680 high performance computing (HPC) jobs in total, using compute nodes in an HPC cluster, which have 5 - 20 assigned physical CPU cores, 30 - 120 GB RAM, and 720 GB local storage. Due to the properties of our resource, we have runtime constraints:

1. Since each HPC job is designed to run for up to 24 hours due to the limited resource, we run a job with a pair of a target tabular dataset and a symbolic regression method.

2. Given a pair of a dataset and a method, each of our HPC jobs runs up to 100 separate training sessions with different hyperparameter values (see Appendix C).

---

15. Since the set of the black-box problems is not either physics-inspired or aligned with our scope of scientific discovery (*e.g.*, car price estimation from car width, height, length, etc), we do not use the datasets in this study.
16. https://chat.openai.com/
17. As of February 6th, 2024, we confirmed using our SRSD Easy set that ChatGPT-4 is not a strong baseline as given a SRSD dataset, it provides Python code to train linear regression models in scikit-learn or SR models in gplearn.
18. For E2E, we used an NVIDIA RTX 3090Ti.

### 5.3 Results

We discuss experimental results of our baseline methods using SRSD datasets. Tables 4 and 5 summarize the performance of the baselines in terms of various metrics for the new datasets without/with dummy variables. With our SRSD datasets, we confirm new findings and a different trend in the overall results compared to those in SRBench as summarized below:

**uDSR and PySR performed the best on our SRSD-Feynman datasets:** According to $R^2$-based accuracy, uDSR significantly outperforms all the other baselines we considered for the *Easy* and *Medium* sets[19], including AIF, which achieved the highest solution rate for the FSRD datasets in SRBench (see Appendix F). Note that $R^2$-based accuracy does not consider the interpretability of the prediction, which is a key property of SR methods and taken into account by solution rate and NED. PySR produced more solutions structurally close to the true models than the other baseline methods and improved the other baseline methods in terms of solution rate and NED. The results of uDSR and PySR also indicate difficulty levels of the three categories of our SRSD datasets, which looks aligned with our complexity-aware dataset categorization (Section 3.2.2).

**None of the baseline methods is robust against dummy variables:** Overall, the performance differences between SRSD-Feynman datasets without and with dummy variables in Tables 4 and 5 highlight that dummy variables made the SRSD problems even more challenging. The dummy variable usage in Table 6 indicates that all the baseline methods considered in this study failed to filter out random dummy variables, which the true models do not use and thus predicted solutions should not use. It should be notable that while PySR's overall performance degraded due to the dummy variables, none of the PySR's solutions that include at least one dummy variable does not achieve $R^2 > 0.999$. Even for SRSD datasets with dummy variables, PySR performed best among the considered baseline methods in terms of NED. Those two trends suggest that it may be important in SRSD problems to penalize overcomplex solutions in a similar way to PySR (Cranmer, 2023).

**$R^2$-based accuracy is vulnerable to dummy variables:** Table 5 shows that compared to the results for SRSD-Feynman (Table 4), AFP and DSR achieved comparable or even improved $R^2$-based accuracy for the datasets with dummy variables. However, the results do not necessarily mean that those methods successfully filter out dummy variables. While DSR performed incredibly better than other baseline methods in terms of $R^2$-based accuracy on the datasets with dummy variables, approximately 45.1% of its non-zero predicted equations that meet $R^2 > 0.999$ ("correct") use at least one dummy variable (Table 6 (right)). Similarly, 100% of the "correct" equations from E2E meet the same conditions.

**NED provides a more fine-grained analysis than solution rate does:** As pointed out in Section 4.1, solution rate is based on binary evaluations[10] and does not help us how structurally close the predicted equation is to the true equation. Thus, the solution rate may be not informative for challenging datasets, which is highlighted at the sparse rows of "Solution Rate" for SRSD-Feynman + Dummy Variables (especially Medium and Hard sets) in Table 5. As demonstrated in the table, NED overcomes the drawback of solution rate and enables comparisons between AFP, AFP-FE, and DSR for the datasets with dummy variables while solution rate shows 0.00% for most of the configurations.

---

19. Chi-squared tests for uDSR and PySR (the second best method in terms of $R^2$-based accuracy) showed p-values of $1.08 \times 10^{-4}$ and $1.37 \times 10^{-4}$, respectively.

Table 4: Baseline results for SRSD-Feynman from various perspectives: 1) accuracy ($R^2 >$ 0.999) (La Cava et al., 2021), 2) solution rate (La Cava et al., 2021), and 3) NED (normalized edit distance).

| Metric | Group | SRSD-Feynman | | | | | | | |
|---|---|---|---|---|---|---|---|---|---|
| | | gplearn | AFP | AFP-FE | AIF | DSR | E2E | uDSR | PySR |
| Accuracy $R^2 > 0.999$ | Easy | 6.67% | 20.0% | 26.7% | 33.3% | 63.3% | 26.7% | **100.0%** | 66.7% |
| | Medium | 7.50% | 2.50% | 2.50% | 5.00% | 45.0% | 17.5% | **75.0%** | 45.0% |
| | Hard | 2.00% | 4.00% | 4.00% | 6.00% | 28.0% | 14.0% | 20.0% | **38.0%** |
| Solution Rate | Easy | 6.67% | 20.0% | 23.3% | 30.0% | 46.7% | 0.00% | 50.0% | **60.0%** |
| | Medium | 0.00% | 2.50% | 2.50% | 2.50% | 10.0% | 0.00% | 17.5% | **30.0%** |
| | Hard | 0.00% | 0.00% | 0.00% | 2.00% | 2.00% | 0.00% | **4.00%** | **4.00%** |
| NED | Easy | 0.866 | 0.727 | 0.693 | 0.646 | 0.524 | 1.00 | 0.478 | **0.269** |
| | Medium | 0.917 | 0.873 | 0.897 | 0.936 | 0.793 | 1.00 | 0.781 | **0.537** |
| | Hard | 0.968 | 0.946 | 0.954 | 0.927 | 0.839 | 0.987 | 0.949 | **0.785** |

Table 5: Baseline results for SRSD-Feynman with dummy variables from various perspectives: 1) accuracy ($R^2 > 0.999$) (La Cava et al., 2021), 2) solution rate (La Cava et al., 2021), and 3) NED (normalized edit distance).

| Metric | Group | SRSD-Feynman + Dummy Variables | | | | | | | |
|---|---|---|---|---|---|---|---|---|---|
| | | gplearn | AFP | AFP-FE | AIF | DSR | E2E | uDSR | PySR |
| Accuracy $R^2 > 0.999$ | Easy | 0.00% | 20.0% | 16.7% | 6.67% | **76.7%** | 16.7% | 53.3% | 20.0% |
| | Medium | 0.00% | 5.00% | 0.00% | 0.00% | **45.0%** | 12.5% | 37.5% | 10.0% |
| | Hard | 0.00% | 4.00% | 4.00% | 0.00% | **22.0%** | 10.0% | 12.0% | 2.00% |
| Solution Rate | Easy | 0.00% | 16.7% | 16.7% | 0.00% | 10.0% | 0.00% | 10.0% | **20.0%** |
| | Medium | 0.00% | 0.00% | 0.00% | 0.00% | 0.00% | 0.00% | **7.50%** | 5.00% |
| | Hard | 0.00% | 0.00% | 0.00% | 0.00% | **2.00%** | 0.00% | 0.00% | 0.00% |
| NED | Easy | 0.963 | 0.769 | 0.786 | 0.975 | 0.771 | 1.00 | 0.871 | **0.418** |
| | Medium | 0.978 | 0.932 | 0.935 | 1.00 | 0.841 | 1.00 | 0.916 | **0.625** |
| | Hard | 0.989 | 0.961 | 0.963 | 1.00 | **0.800** | 1.00 | 0.967 | 0.819 |

## 6 User Study: $R^2$ Score & NED

To investigate how aligned with human judges the existing SR and new SRSD evaluation metrics are, we recruited 23 volunteers from industry and academia who either have doctoral degrees (scientists, professors, engineers) or are doctoral students, and performed a user study with approval from an ethics review board. The volunteers are in diverse research

Table 6: Percentages of predictions that use at least one dummy variable (left) and those also considered "correct" as $R^2 > 0.999$ (right). N/A: Denominator is zero.

| Group | $\geq 1$ dummy variable used | | | | | | | |
|---|---|---|---|---|---|---|---|---|
| | gplearn | AFP | AFP-FE | AIF | DSR | E2E | uDSR | PySR |
| Easy | 66.7% | 52.9% | 56.3% | 50.0% | 53.3% | 100% | 75.0% | 63.3% |
| Medium | 0.00% | 70.6% | 43.8% | N/A | 59.0% | 100% | 66.7% | 67.5% |
| Hard | 100.0% | 81.8% | 58.3% | N/A | 56.3% | 100% | 46.7% | 64.0% |

| Group | $\geq 1$ dummy variable used & $R^2 > 0.999$ | | | | | | | |
|---|---|---|---|---|---|---|---|---|
| | gplearn | AFP | AFP-FE | AIF | DSR | E2E | uDSR | PySR |
| Easy | N/A | 16.7% | 0.00% | 50.0% | 47.8% | 100% | 75.0% | 0.00% |
| Medium | N/A | 50.0% | N/A | N/A | 44.4% | 100% | 73.3% | 0.00% |
| Hard | N/A | 50.0% | 0.00% | N/A | 36.4% | 100% | 33.3% | 0.00% |

fields such as computer science, mathematics, physics, chemistry, material science, aerospace engineering, engineering, medical nutrition, and computational biology. Given a pair of true and estimate equations for an SRSD problem, the subjects were asked to assess an estimated equation on a discretized 1-to-5 scale, where 1 and 5 indicate "1: Completely different from the true equation" and "5: Equivalent to the true equation" respectively. We chose SRSD problems among the 120 SRSD datasets such that we can obtain from the experimental results in Section 5 at least two different equations estimated by different methods that are best among the baseline methods in terms of $R^2$ score and normalized edit distance (NED), respectively. There were 24 resulting SRSD problems for the user study. Note that we do not consider solution rate in the user study. Different from $R^2$ score and NED (real values), we get a binary score (0 or 1) per predicted equation for solution rate, and the best predicted equation in terms of the binary score will be identical to the best predicted equation in terms of NED or randomly chosen from all the predicted equations if the best binary score is 0 *i.e.*, none of the predicted equations is identical to the true equation.

Table 7 shows Pearson correlation coefficients (PCCs) between the human judges and SR/SRSD evaluation metrics. For normalized edit distance (NED), the Pearson correlation coefficient and p-value were $-0.416$ and $1.85 \times 10^{-24}$ respectively, which show a much stronger and statistically more significant correlation between NED and human judges than one for $R^2$ scores. In other words, the results suggest that normalized

Table 7: Pearson correlation coefficients (PCCs) between the human judges and SR/SRSD metrics.

| Metrics | PCC | P-value |
|---|---|---|
| SR: $R^2$ score | $4.66 \times 10^{-3}$ | 0.913 |
| SRSD: NED | $-0.416$ | $1.85 \times 10^{-24}$ |

edit distance is more aligned with human judges than $R^2$ score, and thus can be a better estimate of how close to the true equations the estimated equations are, in a more human-understandable way. Note that the smaller an NED is, the better (structurally closer to the true model) the solution is, thus the negative correlation coefficient is expected.

## 7 Conclusion

In this work, we pointed out issues of existing datasets (*e.g.*, FSRD) and benchmarks of symbolic regression for scientific discovery (SRSD). To address the issues, we proposed 1) 120 new SRSD datasets based on a set of physics formulas in FSRD (Udrescu and Tegmark, 2020), 2) another 120 new SRSD datasets containing dummy variables, and 3) a new evaluation metric for SRSD to discuss the structural similarity between the true and estimated symbolic expressions (equations). The benchmark results revealed key findings including uDSR and PySR being the state of the art for the SRSD datasets (AIF performed the best for the FSRD datasets in SRBench (La Cava et al., 2021)) and the vulnerability of both the SR baselines and $R^2$-based accuracy to dummy variables in SRSD problems. The experimental results and user study also suggest that the normalized edit distance is an additional reasonable metric for SRSD, which provides a more fine-grained analysis than solution rate and incorporates existing SR metrics (*e.g.*, Tables 4 and 5). We also summarize the limitations of this work in Appendix G. Last but not least, the experimental results suggest that our SRSD datasets are more challenging for existing SR methods than the FSRD datasets, and there is still significant room for improvement in SR methods for SRSD problems according to the metrics considered in this study. To encourage the studies of SRSD, we publish our datasets and code with Creative Commons Attribution 4.0 (CC BY) and MIT License, respectively.

## Broader Impact Statement

Intended uses of our proposed datasets and evaluation criteria are for scientific knowledge discoveries such as hidden laws in physics. The datasets consist of physics formulas and do not include any social/personal information. We believe that a potential positive societal consequence of this work is that our benchmark datasets and evaluation criteria will help non-ML experts choose which symbolic regression (SR) methods they want to apply to their problem for scientific discoveries. The proposed datasets and evaluation criteria are more realistic for discussing scientific discovery than existing SR datasets and evaluation criteria as 1) we carefully reviewed the properties of the physics formulas and designed new annotation policies for the proposed datasets, 2) we introduced dummy variables to the proposed datasets to discuss the robustness of SR methods, and 3) our user study shows that the proposed NED is more aligned with human judges than a popular existing SR metric. There are also important considerations regarding the benchmark. When applying SR method to real-world problems, there must be observation noises. However, simulating such realistic noise injections for creating datasets is a challenging task, and our benchmark might bias research in favor of methods that work well for problems where such realistic noises do not exist. Instead, we introduced an assumption that the observed data may include dummy variables, which are not necessary to explain a hidden law.

## Acknowledgments and Disclosure of Funding

This work was supported by JST-Mirai Program Grant Number JPMJMI21G2, JST Moonshot R&D Program Grant Number JPMJMS2236, and JSPS KAKENHI Grant Number 21K14130, Japan. We used computational resource of AI Bridging Cloud Infrastructure (ABCI) provided by National Institute of Advanced Industrial Science and Technology (AIST).

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

## Appendix A. Our SRSD Datasets: Additional Information

This section provides additional information about our SRSD datasets. We created the datasets to discuss the performance of symbolic regression for scientific discovery (SRSD). Each of the 120 SRSD datasets consists of 10,000 samples and has train, val, and test splits with ratio of 8:1:1. For the annotation policy of our SRSD datasets, we refer readers to Section 3.2.1.

Tables 8 – 18 comprehensively summarize the differences between FSRD and our SRSD datasets. Note that the table of Easy set (part 1) is provided as Table 2 in Section 3.1. As described in Section 3.2.2, we categorized each of the 120 SRSD datasets into one of Easy, Medium, and Hard sets. We also introduced dummy variables to the three groups of the datasets, which created another three groups of the datasets (see Section 3.3). We published the six groups of the SRSD datasets with Creative Commons Attribution 4.0 and DOIs (digital object identifiers) at Hugging Face Dataset repositories.The dataset documentations are publicly available as Hugging Face Dataset cards, where we additionally provide an SI derived unit and an SI unit for each of all the constants and variables due to the limited horizontal space in this paper. We also published our codebase as a GitHub repository.[1] These repositories are version-controlled with Git [20] so that users can track the log of the changes. We bear all responsibility in case of violation of rights.

---

20. `https://git-scm.com/`

Table 8: Easy set of our proposed datasets (part 2). C: Constant, V: Variable, F: Float, I: Integer, P: Positive, N: Negative, NN: Non-Negative, I⋆: Integer treated as float due to the capacity of 32-bit integer, $\mathcal{U}$: Uniform distribution, $\mathcal{U}_{\log}$: Log-Uniform distribution.

| Eq. ID | Formula | | Symbols | Properties | | Distributions | |
|---|---|---|---|---|---|---|---|
| | | | | Original | Ours | Original | Ours |
| I.30.5 | $d = \frac{\lambda}{n\sin\theta}$ | $d$ | Interplanar distance | V, F, P | V, F, P | $\mathcal{U}(2,5)$ | N/A |
| | | $\lambda$ | Wavelength of X-ray | V, F, P | V, F, P | $\mathcal{U}(1,2)$ | $\mathcal{U}_{\log}(10^{-11},10^{-9})$ |
| | | $n$ | Number of phase difference | V, F, P | V, I, P | $\mathcal{U}(1,5)$ | $\mathcal{U}_{\log}(10^0,10^2)$ |
| | | $\theta$ | Incidence/Reflection angle | V, F, P | V, F, NN | N/A | $\mathcal{U}(0,\frac{\pi}{2})$ |
| I.43.16 | $v = \mu q \frac{V}{d}$ | $v$ | Velocity | V, F, P | V, F | N/A | N/A |
| | | $\mu$ | Ionic conductivity | V, F, P | V, F | $\mathcal{U}(1,5)$ | $\mathcal{U}_{\log}(10^{-6},10^{-4})$ |
| | | $q$ | Electric charge of ions | V, F, P | V, F | $\mathcal{U}(1,5)$ | $\mathcal{U}_{\log}(10^{-11},10^{-9})$ |
| | | $V$ | Voltage | V, F, P | V, F | $\mathcal{U}(1,5)$ | $\mathcal{U}_{\log}(10^{-1},10^1)$ |
| | | $d$ | Distance | V, F, P | V, F, P | $\mathcal{U}(1,5)$ | $\mathcal{U}_{\log}(10^{-3},10^{-1})$ |
| I.47.23 | $c = \sqrt{\frac{\gamma P}{\rho}}$ | $c$ | Velocity of sound | V, F, P | V, F, P | N/A | N/A |
| | | $\gamma$ | Heat capacity ratio | V, F, P | V, F, P | $\mathcal{U}(1,5)$ | $\mathcal{U}(1,2)$ |
| | | $P$ | Atmospheric pressure | V, F, P | V, F, P | $\mathcal{U}(1,5)$ | $\mathcal{U}(5\times10^{-6},1.5\times10^{-5})$ |
| | | $\rho$ | Density of air | V, F, P | V, F, P | $\mathcal{U}(1,5)$ | $\mathcal{U}(1,2)$ |
| II.2.42 | $J = \kappa(T_2 - T_1)\frac{A}{d}$ | $J$ | Rate of heat flow | V, F | V, F | N/A | N/A |
| | | $\kappa$ | Thermal conductivity | V, F, P | V, F, P | $\mathcal{U}(1,5)$ | $\mathcal{U}_{\log}(10^{-1},10^1)$ |
| | | $T_2$ | Temperature | V, F, P | V, F, P | $\mathcal{U}(1,5)$ | $\mathcal{U}_{\log}(10^1,10^3)$ |
| | | $T_1$ | Temperature | V, F, P | V, F, P | $\mathcal{U}(1,5)$ | $\mathcal{U}_{\log}(10^1,10^3)$ |
| | | $A$ | Area | V, F, P | V, F, P | $\mathcal{U}(1,5)$ | $\mathcal{U}_{\log}(10^{-4},10^{-2})$ |
| | | $d$ | Length | V, F, P | V, F, P | $\mathcal{U}(1,5)$ | $\mathcal{U}_{\log}(10^{-2},10^0)$ |
| II.3.24 | $h = \frac{W}{4\pi r^2}$ | $h$ | Heat flux | V, F, P | V, F | N/A | N/A |
| | | $W$ | Work | V, F, P | V, F | $\mathcal{U}(1,5)$ | $\mathcal{U}_{\log}(10^0,10^2)$ |
| | | $r$ | Distance | V, F, P | V, F, P | $\mathcal{U}(1,5)$ | $\mathcal{U}_{\log}(10^{-2},10^0)$ |
| II.4.23 | $\phi = \frac{q}{4\pi\epsilon r}$ | $\phi$ | Electric potential | V, F, P | V, F | N/A | N/A |
| | | $q$ | Electric charge | V, F, P | V, F | $\mathcal{U}(1,5)$ | $\mathcal{U}_{\log}(10^{-3},10^{-1})$ |
| | | $\epsilon$ | Vacuum permittivity | V, F, P | C, F, P | $\mathcal{U}(1,5)$ | $8.854\times10^{-12}$ |
| | | $r$ | Distance | V, F, P | V, F, P | $\mathcal{U}(1,5)$ | $\mathcal{U}_{\log}(10^{-2},10^0)$ |
| II.8.31 | $u = \frac{\epsilon E^2}{2}$ | $u$ | Energy | V, F, P | V, F, P | N/A | N/A |
| | | $\epsilon$ | Vacuum permittivity | V, F, P | C, F, P | $\mathcal{U}(1,5)$ | $8.854\times10^{-12}$ |
| | | $E$ | Magnitude of electric field | V, F, P | V, F, P | $\mathcal{U}(1,5)$ | $\mathcal{U}_{\log}(10^1,10^3)$ |
| II.10.9 | $E = \frac{\sigma_{\text{free}}}{\epsilon}\frac{1}{1+\chi}$ | $E$ | Electric field | V, F, P | V, F | N/A | N/A |
| | | $\sigma_{\text{free}}$ | Surface charge | V, F, P | V, F | $\mathcal{U}(1,5)$ | $\mathcal{U}_{\log}(10^{-3},10^{-1})$ |
| | | $\epsilon$ | Vacuum permittivity | V, F, P | C, F, P | $\mathcal{U}(1,5)$ | $8.854\times10^{-12}$ |
| | | $\chi$ | Electric susceptibility | V, F, P | V, F, P | $\mathcal{U}(1,5)$ | $\mathcal{U}_{\log}(10^0,10^2)$ |
| II.13.17 | $B = \frac{1}{4\pi\epsilon c^2}\frac{2I}{r}$ | $B$ | Magnitude of the magnetic field | V, F, P | V, F | N/A | N/A |
| | | $\epsilon$ | Vacuum permittivity | V, F, P | C, F, P | $\mathcal{U}(1,5)$ | $8.854\times10^{-12}$ |
| | | $c$ | Speed of light | V, F, P | C, F, P | $\mathcal{U}(1,5)$ | $2.998\times10^8$ |
| | | $I$ | Electric current | V, F, P | V, F | $\mathcal{U}(1,5)$ | $\mathcal{U}_{\log}(10^{-3},10^{-1})$ |
| | | $r$ | Radius | V, F, P | V, F, P | $\mathcal{U}(1,5)$ | $\mathcal{U}_{\log}(10^{-3},10^{-1})$ |
| II.15.4 | $U = -\mu B\cos\theta$ | $U$ | Energy from magnetic field | V, F | V, F | N/A | N/A |
| | | $\mu$ | Magnetic dipole moment | V, F, P | V, F | $\mathcal{U}(1,5)$ | $\mathcal{U}_{\log}(10^{-25},10^{-23})$ |
| | | $B$ | Magnetic field strength | V, F, P | V, F | $\mathcal{U}(1,5)$ | $\mathcal{U}_{\log}(10^{-3},10^{-1})$ |
| | | $\theta$ | Angle | V, F, P | V, F, NN | $\mathcal{U}(1,5)$ | $\mathcal{U}(0,2\pi)$ |

Table 9: Easy set of our proposed datasets (part 3).

| Eq. ID | Formula | | Symbols | Properties | | Distributions | |
|---|---|---|---|---|---|---|---|
| | | | | Original | Ours | Original | Ours |
| II.15.5 | $U = -pE\cos\theta$ | $U$ | Energy | V, F | V, F | N/A | N/A |
| | | $p$ | Electric dipole moment | V, F, P | V, F | $\mathcal{U}(1,5)$ | $\mathcal{U}_{\log}(10^{-22}, 10^{-20})$ |
| | | $E$ | Magnitude of electric field | V, F, P | V, F | $\mathcal{U}(1,5)$ | $\mathcal{U}_{\log}(10^{1}, 10^{3})$ |
| | | $\theta$ | Angle | V, F, P | V, F, NN | $\mathcal{U}(1,5)$ | $\mathcal{U}(0, 2\pi)$ |
| II.27.16 | $L = \epsilon c E^2$ | $L$ | Radiance | V, F, P | V, F | N/A | N/A |
| | | $\epsilon$ | Vacuum permittivity | V, F, P | C, F, P | $\mathcal{U}(1,5)$ | $8.854 \times 10^{-12}$ |
| | | $c$ | Speed of light | V, F, P | C, F, P | $\mathcal{U}(1,5)$ | $2.998 \times 10^{8}$ |
| | | $E$ | Magnitude of electric field | V, F, P | V, F, P | $\mathcal{U}(1,5)$ | $\mathcal{U}_{\log}(10^{-1}, 10^{1})$ |
| II.27.18 | $u = \epsilon E^2$ | $u$ | Energy density | V, F, P | V, F, P | N/A | N/A |
| | | $\epsilon$ | Vacuum permittivity | V, F, P | C, F, P | $\mathcal{U}(1,5)$ | $8.854 \times 10^{-12}$ |
| | | $E$ | Magnitude of electric field | V, F, P | V, F, P | $\mathcal{U}(1,5)$ | $\mathcal{U}_{\log}(10^{-1}, 10^{1})$ |
| II.34.11 | $\omega = g\frac{qB}{2m}$ | $\omega$ | Angular frequency | V, F, P | V, F, P | N/A | N/A |
| | | $g$ | g-factor | V, F, P | V, F | $\mathcal{U}(1,5)$ | $\mathcal{U}(-1, 1)$ |
| | | $q$ | Electric charge | V, F, P | V, F | $\mathcal{U}(1,5)$ | $\mathcal{U}_{\log}(10^{-11}, 10^{-9})$ |
| | | $B$ | Magnetic field strength | V, F, P | V, F | $\mathcal{U}(1,5)$ | $\mathcal{U}_{\log}(10^{-9}, 10^{-7})$ |
| | | $m$ | Mass | V, F, P | V, F, P | $\mathcal{U}(1,5)$ | $\mathcal{U}_{\log}(10^{-30}, 10^{-28})$ |
| II.34.29b | $U = 2\pi g\mu B\frac{J_z}{h}$ | $U$ | Energy | V, F, P | V, F | N/A | N/A |
| | | $g$ | g-factor | V, F, P | V, F | $\mathcal{U}(1,5)$ | $\mathcal{U}(-1, 1)$ |
| | | $\mu$ | Bohr magneton | V, F, P | C, F, P | $\mathcal{U}(1,5)$ | $9.2740100783 \times 10^{-24}$ |
| | | $B$ | Magnetic field strength | V, F, P | V, F | $\mathcal{U}(1,5)$ | $\mathcal{U}_{\log}(10^{-3}, 10^{-1})$ |
| | | $J_z$ | Element of angular momentum | V, F, P | V, F | $\mathcal{U}(1,5)$ | $\mathcal{U}_{\log}(10^{-26}, 10^{-22})$ |
| | | $h$ | Planck constant | V, F, P | C, F, P | $\mathcal{U}(1,5)$ | $6.626 \times 10^{-34}$ |
| II.38.3 | $F = YA\frac{\Delta l}{l}$ | $F$ | Force | V, F, P | V, F | N/A | N/A |
| | | $Y$ | Young's modulus | V, F, P | V, F, P | $\mathcal{U}(1,5)$ | $\mathcal{U}_{\log}(10^{-1}, 10^{1})$ |
| | | $A$ | Area | V, F, P | V, F, P | $\mathcal{U}(1,5)$ | $\mathcal{U}_{\log}(10^{-4}, 10^{-2})$ |
| | | $\delta l$ | Displacement | V, F, P | V, F | $\mathcal{U}(1,5)$ | $\mathcal{U}_{\log}(10^{-3}, 10^{-1})$ |
| | | $l$ | Length | V, F, P | V, F, P | $\mathcal{U}(1,5)$ | $\mathcal{U}_{\log}(10^{-2}, 10^{0})$ |
| II.38.14 | $\mu = \frac{Y}{2(1+\sigma)}$ | $\mu$ | Rigidity modulus | V, F, P | V, F, P | N/A | N/A |
| | | $Y$ | Young's modulus | V, F, P | V, F, P | $\mathcal{U}(1,5)$ | $\mathcal{U}_{\log}(10^{-1}, 10^{1})$ |
| | | $\sigma$ | Poisson coefficient | V, F, P | V, F, P | $\mathcal{U}(1,5)$ | $\mathcal{U}_{\log}(10^{-2}, 10^{0})$ |
| III.7.38 | $\omega = \frac{4\pi\mu B}{h}$ | $\omega$ | Precession frequency | V, F, P | V, F | N/A | N/A |
| | | $\mu$ | Magnetic moment | V, F, P | V, F | $\mathcal{U}(1,5)$ | $\mathcal{U}_{\log}(10^{-11}, 10^{-9})$ |
| | | $B$ | Magnetic flux density | V, F, P | V, F | $\mathcal{U}(1,5)$ | $\mathcal{U}_{\log}(10^{-3}, 10^{-1})$ |
| | | $h$ | Planck constant | V, F, P | C, F, P | $\mathcal{U}(1,5)$ | $6.626 \times 10^{-34}$ |
| III.12.43 | $J = \frac{mh}{2\pi}$ | $J$ | Spin magnetic moment | V, F, P | V, F, P | N/A | N/A |
| | | $m$ | Spin state | V, F, P | V, I, NN | $\mathcal{U}(1,5)$ | $\mathcal{U}_{\log}(10^{0}, 10^{2})$ |
| | | $h$ | Planck constant | V, F, P | C, F, P | $\mathcal{U}(1,5)$ | $6.626 \times 10^{-34}$ |
| III.15.27 | $k = \frac{2\pi}{Nb}s$ | $k$ | Wavenumber | V, F, P | V, F | N/A | N/A |
| | | $s$ | Parameter of state | V, F, P | V, I | $\mathcal{U}(1,5)$ | $\mathcal{U}_{\log}(10^{0}, 10^{2})$ |
| | | $N$ | Number of atoms | V, F, P | V, I, P | $\mathcal{U}(1,5)$ | $\mathcal{U}_{\log}(10^{0}, 10^{2})$ |
| | | $b$ | Lattice constant | V, F, P | V, F, P | $\mathcal{U}(1,5)$ | $\mathcal{U}_{\log}(10^{-10}, 10^{-8})$ |

Table 10: Medium set of our proposed datasets (part 1).

| Eq. ID | Formula | Symbols | | Properties | | Distributions | |
|---|---|---|---|---|---|---|---|
| | | | | Original | Ours | Original | Ours |
| I.8.14 | $d = \sqrt{(x_2 - x_1)^2 + (y_2 - y_1)^2}$ | $d$ | Distance | V, F, P | V, F, NN | N/A | N/A |
| | | $x_2$ | Position | V, F, P | V, F | $\mathcal{U}(1,5)$ | $\mathcal{U}_{\log}(10^{-1}, 10^1)$ |
| | | $x_1$ | Position | V, F, P | V, F | $\mathcal{U}(1,5)$ | $\mathcal{U}_{\log}(10^{-1}, 10^1)$ |
| | | $y_2$ | Position | V, F, P | V, F | $\mathcal{U}(1,5)$ | $\mathcal{U}_{\log}(10^{-1}, 10^1)$ |
| | | $y_1$ | Position | V, F, P | V, F | $\mathcal{U}(1,5)$ | $\mathcal{U}_{\log}(10^{-1}, 10^1)$ |
| I.10.7 | $m = \dfrac{m_0}{\sqrt{1 - \frac{v^2}{c^2}}}$ | $m$ | Mass | V, F, P | V, F, P | N/A | N/A |
| | | $m_0$ | Invariant mass | V, F, P | V, F, P | $\mathcal{U}(1,5)$ | $\mathcal{U}_{\log}(10^{-1}, 10^1)$ |
| | | $v$ | Velocity | V, F, P | V, F, P | $\mathcal{U}(1,2)$ | $\mathcal{U}_{\log}(10^5, 10^8)$ |
| | | $c$ | Speed of light | V, F, P | C, F, P | $\mathcal{U}(3,10)$ | $2.998 \times 10^8$ |
| I.11.19 | $A = x_1 y_1 + x_2 y_2 + x_3 y_3$ | $A$ | Inner product | V, F, P | V, F | N/A | N/A |
| | | $x_1$ | Element of a vector | V, F, P | V, F | $\mathcal{U}(1,5)$ | $\mathcal{U}_{\log}(10^{-1}, 10^1)$ |
| | | $y_1$ | Element of a vector | V, F, P | V, F | $\mathcal{U}(1,5)$ | $\mathcal{U}_{\log}(10^{-1}, 10^1)$ |
| | | $x_2$ | Element of a vector | V, F, P | V, F | $\mathcal{U}(1,5)$ | $\mathcal{U}_{\log}(10^{-1}, 10^1)$ |
| | | $y_2$ | Element of a vector | V, F, P | V, F | $\mathcal{U}(1,5)$ | $\mathcal{U}_{\log}(10^{-1}, 10^1)$ |
| | | $x_3$ | Element of a vector | V, F, P | V, F | $\mathcal{U}(1,5)$ | $\mathcal{U}_{\log}(10^{-1}, 10^1)$ |
| | | $y_3$ | Element of a vector | V, F, P | V, F | $\mathcal{U}(1,5)$ | $\mathcal{U}_{\log}(10^{-1}, 10^1)$ |
| I.12.2 | $F = \dfrac{q_1 q_2}{4\pi\epsilon r^2}$ | $F$ | Electrostatic force | V, F, P | V, F | N/A | N/A |
| | | $q_1$ | Electric charge | V, F, P | V, F | $\mathcal{U}(1,5)$ | $\mathcal{U}_{\log}(10^{-1}, 10^1)$ |
| | | $q_2$ | Electric charge | V, F, P | V, F | $\mathcal{U}(1,5)$ | $\mathcal{U}_{\log}(10^{-1}, 10^1)$ |
| | | $r$ | Distance | V, F, P | V, F, P | $\mathcal{U}(1,5)$ | $\mathcal{U}_{\log}(10^{-1}, 10^1)$ |
| | | $\epsilon$ | Vacuum permittivity | V, F, P | C, F, P | $\mathcal{U}(1,5)$ | $8.854 \times 10^{-12}$ |
| I.12.11 | $F = q\left(E + Bv\sin(\theta)\right)$ | $F$ | Force | V, F | V, F | N/A | N/A |
| | | $q$ | Electric charge | V, F, P | V, F | $\mathcal{U}(1,5)$ | $\mathcal{U}_{\log}(10^{-1}, 10^1)$ |
| | | $E$ | Electric field | V, F, P | V, F, P | $\mathcal{U}(1,5)$ | $\mathcal{U}_{\log}(10^{-1}, 10^1)$ |
| | | $B$ | Magnetic field strength | V, F, P | V, F, P | $\mathcal{U}(1,5)$ | $\mathcal{U}_{\log}(10^{-1}, 10^1)$ |
| | | $v$ | Velocity | V, F, P | V, F, P | $\mathcal{U}(1,5)$ | $\mathcal{U}_{\log}(10^{-1}, 10^1)$ |
| | | $\theta$ | Angle | V, F, P | V, F, NN | $\mathcal{U}(1,5)$ | $\mathcal{U}(0, \frac{\pi}{2})$ |
| I.13.4 | $K = \frac{1}{2}m(v^2 + u^2 + w^2)$ | $K$ | Kinetic energy | V, F, P | V, F, P | N/A | N/A |
| | | $m$ | Mass | V, F, P | V, F, P | $\mathcal{U}(1,5)$ | $\mathcal{U}_{\log}(10^{-1}, 10^1)$ |
| | | $v$ | Element of velocity | V, F, P | V, F | $\mathcal{U}(1,5)$ | $\mathcal{U}_{\log}(10^{-1}, 10^1)$ |
| | | $u$ | Element of velocity | V, F, P | V, F | $\mathcal{U}(1,5)$ | $\mathcal{U}_{\log}(10^{-1}, 10^1)$ |
| | | $w$ | Element of velocity | V, F, P | V, F | $\mathcal{U}(1,5)$ | $\mathcal{U}_{\log}(10^{-1}, 10^1)$ |
| I.13.12 | $U = Gm_1 m_2 \left(\frac{1}{r_2} - \frac{1}{r_1}\right)$ | $U$ | Potential energy | V, F | V, F | N/A | N/A |
| | | $G$ | Gravitational constant | V, F, P | C, F, P | $\mathcal{U}(1,5)$ | $6.674 \times 10^{-11}$ |
| | | $m_1$ | Mass (The Earth) | V, F, P | V, F, P | $\mathcal{U}(1,5)$ | $\mathcal{U}_{\log}(10^{-2}, 10^0)$ |
| | | $m_2$ | Mass | V, F, P | V, F, P | $\mathcal{U}(1,5)$ | $\mathcal{U}_{\log}(10^{-2}, 10^0)$ |
| | | $r_2$ | Distance | V, F, P | V, F, P | $\mathcal{U}(1,5)$ | $\mathcal{U}_{\log}(10^{-2}, 10^0)$ |
| | | $r_1$ | Distance | V, F, P | V, F, P | $\mathcal{U}(1,5)$ | $\mathcal{U}_{\log}(10^{-2}, 10^0)$ |
| I.15.10 | $p = \dfrac{m_0 v}{\sqrt{1 - v^2/c^2}}$ | $p$ | Relativistic momentum | V, F, P | V, F, P | N/A | N/A |
| | | $m_0$ | Rest Mass | V, F, P | V, F, P | $\mathcal{U}(1,5)$ | $\mathcal{U}_{\log}(10^{-2}, 10^0)$ |
| | | $v$ | Velocity | V, F, P | V, F | $\mathcal{U}(1,2)$ | $\mathcal{U}_{\log}(10^5, 10^7)$ |
| | | $c$ | Speed of light | V, F, P | C, F, P | $\mathcal{U}(3,10)$ | $2.998 \times 10^8$ |
| I.16.6 | $v_1 = \dfrac{u+v}{1+uv/c^2}$ | $v_1$ | Velocity | V, F, P | V, F | N/A | N/A |
| | | $u$ | Velocity | V, F, P | V, F | $\mathcal{U}(1,5)$ | $\mathcal{U}_{\log}(10^6, 10^8)$ |
| | | $v$ | Velocity | V, F, P | V, F | $\mathcal{U}(1,5)$ | $\mathcal{U}_{\log}(10^6, 10^8)$ |
| | | $c$ | Speed of light | V, F, P | C, F, P | $\mathcal{U}(1,5)$ | $2.998 \times 10^8$ |
| I.18.4 | $r = \dfrac{m_1 r_1 + m_2 r_2}{m_1 + m_2}$ | $r$ | Center of gravity | V, F, P | V, F | N/A | N/A |
| | | $m_1$ | Mass | V, F, P | V, F, P | $\mathcal{U}(1,5)$ | $\mathcal{U}_{\log}(10^{-1}, 10^1)$ |
| | | $r_1$ | Position | V, F, P | V, F | $\mathcal{U}(1,5)$ | $\mathcal{U}_{\log}(10^{-1}, 10^1)$ |
| | | $m_2$ | Mass | V, F, P | V, F, P | $\mathcal{U}(1,5)$ | $\mathcal{U}_{\log}(10^{-1}, 10^1)$ |
| | | $r_2$ | Position | V, F, P | V, F | $\mathcal{U}(1,5)$ | $\mathcal{U}_{\log}(10^{-1}, 10^1)$ |

Table 11: Medium set of our proposed datasets (part 2).

| Eq. ID | Formula | Symbols | | Properties Original | Properties Ours | Distributions Original | Distributions Ours |
|---|---|---|---|---|---|---|---|
| I.24.6 | $E = \frac{1}{4}m(\omega^2 + \omega_0^2)x^2$ | $E$ | Energy | V, F, P | V, F, P | N/A | N/A |
| | | $m$ | Mass | V, F, P | V, F, P | $\mathcal{U}(1,3)$ | $\mathcal{U}_{\log}(10^{-1}, 10^1)$ |
| | | $\omega$ | Angular velocity | V, F, P | V, F | $\mathcal{U}(1,3)$ | $\mathcal{U}_{\log}(10^{-1}, 10^1)$ |
| | | $\omega_0$ | Angular velocity | V, F, P | V, F | $\mathcal{U}(1,3)$ | $\mathcal{U}_{\log}(10^{-1}, 10^1)$ |
| | | $x$ | Position | V, F, P | V, F | $\mathcal{U}(1,3)$ | $\mathcal{U}_{\log}(10^{-1}, 10^1)$ |
| I.29.4 | $k = \frac{\omega}{c}$ | $k$ | Wavenumber | V, F, P | V, F, P | N/A | N/A |
| | | $\omega$ | Frequency of electromagnetic waves | V, F, P | V, F, P | $\mathcal{U}(1,10)$ | $\mathcal{U}_{\log}(10^9, 10^{11})$ |
| | | $c$ | Speed of light | V, F, P | C, F, P | $\mathcal{U}(1,10)$ | $2.998 \times 10^8$ |
| I.32.5 | $P = \frac{q^2 a^2}{6\pi\epsilon c^3}$ | $P$ | Radiant energy | V, F, P | V, F, P | N/A | N/A |
| | | $q$ | Electric charge | V, F, P | V, F | $\mathcal{U}(1,5)$ | $\mathcal{U}_{\log}(10^{-3}, 10^{-1})$ |
| | | $a$ | Magnitude of direction vector | V, F, P | V, F, P | $\mathcal{U}(1,5)$ | $\mathcal{U}_{\log}(10^5, 10^7)$ |
| | | $\epsilon$ | Vacuum permittivity | V, F, P | C, F, P | $\mathcal{U}(1,5)$ | $8.854 \times 10^{-12}$ |
| | | $c$ | Speed of light | V, F, P | C, F, P | $\mathcal{U}(1,5)$ | $2.998 \times 10^8$ |
| I.34.8 | $\omega = \frac{qvB}{p}$ | $\omega$ | Angular velocity | V, F, P | V, F | N/A | N/A |
| | | $q$ | Electric charge | V, F, P | V, F | $\mathcal{U}(1,5)$ | $\mathcal{U}_{\log}(10^{-11}, 10^{-9})$ |
| | | $v$ | Velocity | V, F, P | V, F | $\mathcal{U}(1,5)$ | $\mathcal{U}_{\log}(10^5, 10^7)$ |
| | | $B$ | Magnetic field | V, F, P | V, F | $\mathcal{U}(1,5)$ | $\mathcal{U}_{\log}(10^1, 10^3)$ |
| | | $p$ | Angular momentum | V, F, P | V, F | $\mathcal{U}(1,5)$ | $\mathcal{U}_{\log}(10^9, 10^{11})$ |
| I.34.10 | $\omega = \frac{\omega_0}{1 - v/c}$ | $\omega$ | Frequency of electromagnetic waves | V, F, P | V, F, P | N/A | N/A |
| | | $\omega_0$ | Frequency of electromagnetic waves | V, F, P | V, F, P | $\mathcal{U}(1,5)$ | $\mathcal{U}_{\log}(10^9, 10^{11})$ |
| | | $v$ | Velocity | V, F, P | V, F | $\mathcal{U}(1,2)$ | $\mathcal{U}_{\log}(10^5, 10^7)$ |
| | | $c$ | Speed of light | V, F, P | C, F, P | $\mathcal{U}(3,10)$ | $2.998 \times 10^8$ |
| I.34.27 | $W = \frac{h}{2\pi}\omega$ | $W$ | Energy | V, F, P | V, F, P | N/A | N/A |
| | | $h$ | Planck constant | V, F, P | C, F, P | $\mathcal{U}(1,5)$ | $6.626 \times 10^{-34}$ |
| | | $\omega$ | Frequency of electromagnetic waves | V, F, P | V, F, P | $\mathcal{U}(1,5)$ | $\mathcal{U}_{\log}(10^9, 10^{11})$ |
| I.38.12 | $r = 4\pi\epsilon \frac{(h/(2\pi))^2}{mq^2}$ | $r$ | Bohr radius | V, F, P | V, F, P | N/A | N/A |
| | | $\epsilon$ | Vacuum permittivity | V, F, P | C, F, P | $\mathcal{U}(1,5)$ | $8.854 \times 10^{-12}$ |
| | | $h$ | Planck constant | V, F, P | C, F, P | $\mathcal{U}(1,5)$ | $6.626 \times 10^{-34}$ |
| | | $m$ | Mass | V, F, P | V, F, P | $\mathcal{U}(1,5)$ | $\mathcal{U}_{\log}(10^{-28}, 10^{-26})$ |
| | | $q$ | Electric charge | V, F, P | V, F, P | $\mathcal{U}(1,5)$ | $\mathcal{U}_{\log}(10^{-11}, 10^{-9})$ |
| I.39.10 | $U = \frac{3}{2}PV$ | $U$ | Internal energy | V, F, P | V, F, P | N/A | N/A |
| | | $P$ | Pressure | V, F, P | V, F, P | $\mathcal{U}(1,5)$ | $\mathcal{U}_{\log}(10^4, 10^6)$ |
| | | $V$ | Volume | V, F, P | V, F, P | $\mathcal{U}(1,5)$ | $\mathcal{U}_{\log}(10^{-5}, 10^{-3})$ |
| I.39.11 | $U = \frac{PV}{\gamma - 1}$ | $U$ | Energy | V, F, P | V, F, P | N/A | N/A |
| | | $\gamma$ | Heat capacity ratio | V, F, P | V, F, P | $\mathcal{U}(2,5)$ | $\mathcal{U}(1,2)$ |
| | | $P$ | Pressure | V, F, P | V, F, P | $\mathcal{U}(1,5)$ | $\mathcal{U}_{\log}(10^4, 10^6)$ |
| | | $V$ | Volume | V, F, P | V, F, P | $\mathcal{U}(1,5)$ | $\mathcal{U}_{\log}(10^{-5}, 10^{-3})$ |
| I.43.31 | $D = \mu kT$ | $D$ | Diffusion coefficient | V, F, P | V, F, P | N/A | N/A |
| | | $\mu$ | Viscosity | V, F, P | V, F, P | $\mathcal{U}(1,5)$ | $\mathcal{U}_{\log}(10^{13}, 10^{15})$ |
| | | $k$ | Boltzmann constant | V, F, P | C, F, P | $\mathcal{U}(1,5)$ | $1.381 \times 10^{-23}$ |
| | | $T$ | Temperature | V, F, P | V, F, P | $\mathcal{U}(1,5)$ | $\mathcal{U}_{\log}(10^1, 10^3)$ |

Table 12: Medium set of our proposed datasets (part 3).

| Eq. ID | Formula | Symbols | | Properties | | Distributions | |
|---|---|---|---|---|---|---|---|
| | | | | Original | Ours | Original | Ours |
| I.43.43 | $\kappa = \frac{1}{\gamma-1}\frac{kv}{\sigma_c}$ | $\kappa$ | Thermal conductivity | V, F, P | V, F, P | N/A | N/A |
| | | $\gamma$ | Heat capacity ratio | V, F, P | V, F, P | $\mathcal{U}(2,5)$ | $\mathcal{U}(1,2)$ |
| | | $k$ | Boltzmann constant | V, F, P | C, F, P | $\mathcal{U}(1,5)$ | $1.381 \times 10^{-23}$ |
| | | $v$ | Velocity | V, F, P | V, F, P | $\mathcal{U}(1,5)$ | $\mathcal{U}_{\log}(10^2,10^4)$ |
| | | $\sigma_c$ | Molecular collision cross section | V, F, P | V, F, P | $\mathcal{U}(1,5)$ | $\mathcal{U}_{\log}(10^{-21},10^{-19})$ |
| I.48.2 | $E = \frac{mc^2}{\sqrt{1-v^2/c^2}}$ | $E$ | Energy | V, F, P | V, F, P | N/A | N/A |
| | | $m$ | Mass | V, F, P | V, F, P | $\mathcal{U}(1,5)$ | $\mathcal{U}_{\log}(10^{-29},10^{-27})$ |
| | | $c$ | Speed of light | V, F, P | C, F, P | $\mathcal{U}(3,10)$ | $2.998 \times 10^8$ |
| | | $v$ | Velocity | V, F, P | V, F, P | $\mathcal{U}(1,2)$ | $\mathcal{U}_{\log}(10^6,10^8)$ |
| II.6.11 | $\phi = \frac{1}{4\pi\epsilon}\frac{p\cos\theta}{r^2}$ | $\phi$ | Electric potential | V, F | V, F | N/A | N/A |
| | | $\epsilon$ | Vacuum permittivity | V, F, P | C, F, P | $\mathcal{U}(1,3)$ | $8.854 \times 10^{-12}$ |
| | | $p$ | Electric dipole moment | V, F, P | V, F | $\mathcal{U}(1,3)$ | $\mathcal{U}_{\log}(10^{-22},10^{-20})$ |
| | | $\theta$ | Angle | V, F, P | V, F, NN | $\mathcal{U}(1,3)$ | $\mathcal{U}(0,2\pi)$ |
| | | $r$ | Distance | V, F, P | V, F, P | $\mathcal{U}(1,3)$ | $\mathcal{U}_{\log}(10^{-10},10^{-8})$ |
| II.8.7 | $U = \frac{3}{5}\frac{Q^2}{4\pi\epsilon a}$ | $U$ | Energy | V, F, P | V, F, P | N/A | N/A |
| | | $Q$ | Electric charge | V, F, P | V, F | $\mathcal{U}(1,5)$ | $\mathcal{U}_{\log}(10^{-11},10^{-9})$ |
| | | $\epsilon$ | Vacuum permittivity | V, F, P | C, F, P | $\mathcal{U}(1,5)$ | $8.854 \times 10^{-12}$ |
| | | $a$ | Radius | V, F, P | V, F, P | $\mathcal{U}(1,5)$ | $\mathcal{U}_{\log}(10^{-12},10^{-10})$ |
| II.11.3 | $x = \frac{qE}{m(\omega_0^2-\omega^2)}$ | $x$ | Position | V, F, P | V, F | N/A | N/A |
| | | $q$ | Electric charge | V, F, P | V, F | $\mathcal{U}(1,3)$ | $\mathcal{U}_{\log}(10^{-11},10^{-9})$ |
| | | $E$ | Magnitude of electric field | V, F, P | V, F, P | $\mathcal{U}(1,3)$ | $\mathcal{U}_{\log}(10^{-9},10^{-7})$ |
| | | $m$ | Mass | V, F, P | V, F, P | $\mathcal{U}(1,3)$ | $\mathcal{U}_{\log}(10^{-28},10^{-26})$ |
| | | $\omega_0$ | Angular velocity | V, F, P | V, F | $\mathcal{U}(3,5)$ | $\mathcal{U}_{\log}(10^9,10^{11})$ |
| | | $\omega$ | Angular velocity | V, F, P | V, F | $\mathcal{U}(1,2)$ | $\mathcal{U}_{\log}(10^9,10^{11})$ |
| II.21.32 | $\phi = \frac{q}{4\pi\epsilon r(1-v/c)}$ | $\phi$ | Electric potential | V, F, P | V, F | N/A | N/A |
| | | $q$ | Electric charge | V, F, P | V, F | $\mathcal{U}(1,5)$ | $\mathcal{U}_{\log}(10^{-3},10^{-1})$ |
| | | $\epsilon$ | Vacuum permittivity | V, F, P | C, F, P | $\mathcal{U}(1,5)$ | $8.854 \times 10^{-12}$ |
| | | $r$ | Distance | V, F, P | V, F, P | $\mathcal{U}(1,5)$ | $\mathcal{U}_{\log}(10^0,10^2)$ |
| | | $v$ | Velocity | V, F, P | V, F, P | $\mathcal{U}(1,2)$ | $\mathcal{U}_{\log}(10^6,10^8)$ |
| | | $c$ | Speed of light | V, F, P | C, F, P | $\mathcal{U}(3,10)$ | $2.998 \times 10^8$ |
| II.34.2 | $\mu = \frac{qvr}{2}$ | $\mu$ | Magnetic moment | V, F, P | V, F | N/A | N/A |
| | | $q$ | Electric charge | V, F, P | V, F | $\mathcal{U}(1,5)$ | $\mathcal{U}_{\log}(10^{-11},10^{-9})$ |
| | | $v$ | Velocity | V, F, P | V, F | $\mathcal{U}(1,5)$ | $\mathcal{U}_{\log}(10^5,10^7)$ |
| | | $r$ | Radius | V, F, P | V, F, P | $\mathcal{U}(1,5)$ | $\mathcal{U}_{\log}(10^{-11},10^{-9})$ |
| II.34.2a | $I = \frac{qv}{2\pi r}$ | $I$ | Electric Current | V, F, P | V, F | N/A | N/A |
| | | $q$ | Electric charge | V, F, P | V, F | $\mathcal{U}(1,5)$ | $\mathcal{U}_{\log}(10^{-11},10^{-9})$ |
| | | $v$ | Velocity | V, F, P | V, F | $\mathcal{U}(1,5)$ | $\mathcal{U}_{\log}(10^5,10^7)$ |
| | | $r$ | Radius | V, F, P | V, F, P | $\mathcal{U}(1,5)$ | $\mathcal{U}_{\log}(10^{-11},10^{-9})$ |
| II.34.29a | $\mu = \frac{qh}{4\pi m}$ | $\mu$ | Bohr magneton | V, F, P | V, F | N/A | N/A |
| | | $q$ | Electric charge | V, F, P | V, F | $\mathcal{U}(1,5)$ | $\mathcal{U}_{\log}(10^{-11},10^{-9})$ |
| | | $h$ | Planck constant | V, F, P | C, F, P | $\mathcal{U}(1,5)$ | $6.626 \times 10^{-34}$ |
| | | $m$ | Mass | V, F, P | V, F, P | $\mathcal{U}(1,5)$ | $\mathcal{U}_{\log}(10^{-30},10^{-28})$ |
| II.37.1 | $E = \mu(1+\chi)B$ | $E$ | Energy of magnetic field | V, F, P | V, F | N/A | N/A |
| | | $\mu$ | Magnetic moment | V, F, P | V, F | $\mathcal{U}(1,5)$ | $\mathcal{U}_{\log}(10^{-25},10^{-23})$ |
| | | $\chi$ | Volume magnetic susceptibility | V, F, P | V, F | $\mathcal{U}(1,5)$ | $\mathcal{U}_{\log}(10^4,10^6)$ |
| | | $B$ | Magnetic field strength | V, F, P | V, F | $\mathcal{U}(1,5)$ | $\mathcal{U}_{\log}(10^{-3},10^{-1})$ |

Table 13: Medium set of our proposed datasets (part 4).

| Eq. ID | Formula | | Symbols | Properties Original | Ours | Distributions Original | Ours |
|---|---|---|---|---|---|---|---|
| III.4.32 | $n = \frac{1}{\exp(h\omega/2\pi kT)-1}$ | $n$ | Average number of photons | V, F, P | V, F, P | N/A | N/A |
| | | $h$ | Planck constant | V, F, P | C, F, P | $\mathcal{U}(1,5)$ | $6.626 \times 10^{-34}$ |
| | | $\omega$ | Frequency | V, F, P | V, F, P | $\mathcal{U}(1,5)$ | $\mathcal{U}_{\log}(10^8, 10^{10})$ |
| | | $k$ | Boltzmann constant | V, F, P | C, F, P | $\mathcal{U}(1,5)$ | $1.381 \times 10^{-23}$ |
| | | $T$ | Temperature | V, F, P | V, F, P | $\mathcal{U}(1,5)$ | $\mathcal{U}_{\log}(10^1, 10^3)$ |
| III.8.54 | $|C|^2 = \sin^2\left(\frac{2\pi At}{h}\right)$ | $|C|^2$ | Probability | V, F | V, F, NN | N/A | N/A |
| | | $A$ | Energy | V, F, P | V, F | $\mathcal{U}(1,2)$ | $\mathcal{U}_{\log}(10^{-18}, 10^{-16})$ |
| | | $t$ | Time | V, F, P | V, F, NN | $\mathcal{U}(1,2)$ | $\mathcal{U}_{\log}(10^{-18}, 10^{-16})$ |
| | | $h$ | Planck constant | V, F, P | C, F, P | $\mathcal{U}(1,4)$ | $6.626 \times 10^{-34}$ |
| III.13.18 | $v = \frac{4\pi Ab^2}{h}k$ | $v$ | Speed of the waves | V, F, P | V, F | N/A | N/A |
| | | $A$ | Energy | V, F, P | V, F | $\mathcal{U}(1,5)$ | $\mathcal{U}_{\log}(10^{-18}, 10^{-16})$ |
| | | $b$ | Lattice constant | V, F, P | V, F, P | $\mathcal{U}(1,5)$ | $\mathcal{U}_{\log}(10^{-10}, 10^{-8})$ |
| | | $k$ | Wavenumber | V, F, P | V, F, P | $\mathcal{U}(1,5)$ | $\mathcal{U}_{\log}(10^{-1}, 10^1)$ |
| | | $h$ | Planck constant | V, F, P | C, F, P | $\mathcal{U}(1,5)$ | $6.626 \times 10^{-34}$ |
| III.14.14 | $I = I_0\left(\exp\left(q\Delta V/\kappa T\right)-1\right)$ | $I$ | Electric Current | V, F, P | V, F | N/A | N/A |
| | | $I_0$ | Electric current | V, F, P | V, F | $\mathcal{U}(1,5)$ | $\mathcal{U}_{\log}(10^{-3}, 10^{-1})$ |
| | | $q$ | Electric charge | V, F, P | V, F, P | $\mathcal{U}(1,2)$ | $\mathcal{U}_{\log}(10^{-22}, 10^{-20})$ |
| | | $\Delta V$ | Voltage | V, F, P | V, F | $\mathcal{U}(1,2)$ | $\mathcal{U}_{\log}(10^{-1}, 10^1)$ |
| | | $\kappa$ | Boltzmann constant | V, F, P | C, F, P | $\mathcal{U}(1,2)$ | $1.381 \times 10^{-23}$ |
| | | $T$ | Temperature | V, F, P | V, F, P | $\mathcal{U}(1,2)$ | $\mathcal{U}_{\log}(10^1, 10^3)$ |
| III.15.12 | $E = 2A\left(1 - \cos\left(kd\right)\right)$ | $E$ | Energy | V, F | V, F, P | N/A | N/A |
| | | $A$ | Amplitude | V, F, P | V, F, P | $\mathcal{U}(1,5)$ | $\mathcal{U}_{\log}(10^{-18}, 10^{-16})$ |
| | | $k$ | Propagation coefficient | V, F, P | V, F, P | $\mathcal{U}(1,5)$ | $\mathcal{U}_{\log}(10^{-1}, 10^1)$ |
| | | $d$ | Lattice constant | V, F, P | V, F, P | $\mathcal{U}(1,5)$ | $\mathcal{U}_{\log}(10^{-10}, 10^{-8})$ |
| III.15.14 | $m = \frac{h^2}{8\pi^2 Ab^2}$ | $m$ | Effective mass | V, F, P | V, F, P | N/A | N/A |
| | | $h$ | Planck constant | V, F, P | C, F, P | $\mathcal{U}(1,5)$ | $6.626 \times 10^{-34}$ |
| | | $A$ | Amplitude | V, F, P | V, F, P | $\mathcal{U}(1,5)$ | $\mathcal{U}_{\log}(10^{-18}, 10^{-16})$ |
| | | $b$ | Lattice constant | V, F, P | V, F, P | $\mathcal{U}(1,5)$ | $\mathcal{U}_{\log}(10^{-10}, 10^{-8})$ |
| III.17.37 | $f = \beta(1 + \alpha\cos\theta)$ | $f$ | Distribution | V, F | V, F, P | N/A | N/A |
| | | $\beta$ | Variable | V, F, P | V, F, P | $\mathcal{U}(1,5)$ | $\mathcal{U}_{\log}(10^{-18}, 10^{-16})$ |
| | | $\alpha$ | Variable | V, F, P | V, F | $\mathcal{U}(1,5)$ | $\mathcal{U}_{\log}(10^{-18}, 10^{-16})$ |
| | | $\theta$ | Angle | V, F, P | V, F, NN | $\mathcal{U}(1,5)$ | $\mathcal{U}(0, 2\pi)$ |
| III.19.51 | $E = -\frac{mq^4}{2(4\pi\epsilon)^2(h/(2\pi))^2 n^2}$ | $E$ | Energy | V, F, P | V, F, P | N/A | N/A |
| | | $m$ | Mass | V, F, P | V, F, P | $\mathcal{U}(1,5)$ | $\mathcal{U}_{\log}(10^{-30}, 10^{-28})$ |
| | | $q$ | Electric charge | V, F, P | V, F | $\mathcal{U}(1,5)$ | $\mathcal{U}_{\log}(10^{-11}, 10^{-9})$ |
| | | $\epsilon$ | Vacuum permittivity | V, F, P | C, F, P | $\mathcal{U}(1,5)$ | $8.854 \times 10^{-12}$ |
| | | $h$ | Planck constant | V, F, P | C, F, P | $\mathcal{U}(1,5)$ | $6.626 \times 10^{-34}$ |
| | | $n$ | Number of protons | V, F, P | V, I, P | $\mathcal{U}(1,5)$ | $\mathcal{U}_{\log}(10^0, 10^2)$ |
| B8 | $U = \frac{E}{1+\frac{E}{mc^2}(1-\cos\theta)}$ | $U$ | Energy | V, F, P | V, F, P | N/A | N/A |
| | | $E$ | Electromagnetic energy | V, F, P | V, F, P | $\mathcal{U}(1,3)$ | $\mathcal{U}_{\log}(10^{-24}, 10^{-22})$ |
| | | $m$ | Electron mass | V, F, P | C, F, P | $\mathcal{U}(1,3)$ | $9.109 \times 10^{-31}$ |
| | | $c$ | Speed of light | V, F, P | C, F, P | $\mathcal{U}(1,3)$ | $2.998 \times 10^8$ |
| | | $\theta$ | Incidence angle | V, F, P | V, F | $\mathcal{U}(1,3)$ | $\mathcal{U}(-\pi, \pi)$ |
| B18 | $\rho = \frac{3}{8\pi G}\left(\frac{c^2 k_{\mathrm{f}}}{a_{\mathrm{f}}^2} + H^2\right)$ | $\rho$ | Density | V, F, P | V, F | N/A | N/A |
| | | $G$ | Gravitational constant | V, F, P | C, F, P | $\mathcal{U}(1,5)$ | $6.674 \times 10^{-11}$ |
| | | $c$ | Speed of light | V, F, P | C, F, P | $\mathcal{U}(1,5)$ | $2.998 \times 10^8$ |
| | | $k_{\mathrm{f}}$ | Curvature of the Universe | V, F, P | V, F | $\mathcal{U}(1,5)$ | $\mathcal{U}_{\log}(10^1, 10^3)$ |
| | | $a_{\mathrm{f}}$ | Distance | V, F, P | V, F, P | $\mathcal{U}(1,5)$ | $\mathcal{U}_{\log}(10^8, 10^{10})$ |
| | | $H$ | Hubble's constant | V, F, P | V, F | $\mathcal{U}(1,5)$ | $\mathcal{U}_{\log}(10^0, 10^2)$ |

Table 14: Hard set of our proposed datasets (part 1).

| Eq. ID | Formula | Symbols | | Properties | | Distributions | |
|---|---|---|---|---|---|---|---|
| | | | | Original | Ours | Original | Ours |
| I.6.20 | $f = \exp\left(-\frac{\theta^2}{2\sigma^2}\right)/\sqrt{2\pi\sigma^2}$ | $f$ | Probability density function | V, F, P | V, F, P | N/A | N/A |
| | | $\theta$ | Position | V, F, P | V, F | $\mathcal{U}(1,3)$ | $\mathcal{U}_{\log}(10^{-1},10^1)$ |
| | | $\sigma$ | Standard deviation | V, F, P | V, F, P | $\mathcal{U}(1,3)$ | $\mathcal{U}_{\log}(10^{-1},10^1)$ |
| I.6.20a | $f = \exp\left(-\frac{\theta^2}{2}\right)/\sqrt{2\pi}$ | $f$ | Probability density function | V, F, P | V, F, P | N/A | N/A |
| | | $\theta$ | Position | V, F, P | V, F | $\mathcal{U}(1,3)$ | $\mathcal{U}_{\log}(10^{-1},10^1)$ |
| I.6.20b | $f = \exp\left(-\frac{(\theta-\theta_1)^2}{2\sigma^2}\right)/\sqrt{2\pi\sigma}$ | $f$ | Probability density function | V, F, P | V, F, P | N/A | N/A |
| | | $\theta$ | Position | V, F, P | V, F | $\mathcal{U}(1,3)$ | $\mathcal{U}_{\log}(10^{-1},10^1)$ |
| | | $\theta_1$ | Position | V, F, P | V, F | $\mathcal{U}(1,3)$ | $\mathcal{U}_{\log}(10^{-1},10^1)$ |
| | | $\sigma$ | Standard deviation | V, F, P | V, F, P | $\mathcal{U}(1,3)$ | $\mathcal{U}_{\log}(10^{-1},10^1)$ |
| I.9.18 | $F = \dfrac{Gm_1 m_2}{(x_2-x_1)^2+(y_2-y_1)^2+(z_2-z_1)^2}$ | $F$ | Force of gravity | V, F, P | V, F, P | N/A | N/A |
| | | $G$ | Gravitational constant | V, F, P | C, F, P | $\mathcal{U}(1,2)$ | $6.674\times10^{-11}$ |
| | | $m_1$ | Mass | V, F, P | V, F, P | $\mathcal{U}(1,2)$ | $\mathcal{U}_{\log}(10^0,10^3)$ |
| | | $m_2$ | Mass | V, F, P | V, F, P | $\mathcal{U}(1,2)$ | $\mathcal{U}_{\log}(10^0,10^3)$ |
| | | $x_2$ | Position | V, F, P | V, F | $\mathcal{U}(1,2)$ | $\mathcal{U}_{\log}(10^0,10^1)$ |
| | | $x_1$ | Position | V, F, P | V, F | $\mathcal{U}(3,4)$ | $\mathcal{U}_{\log}(10^0,10^1)$ |
| | | $y_2$ | Position | V, F, P | V, F | $\mathcal{U}(1,2)$ | $\mathcal{U}_{\log}(10^0,10^1)$ |
| | | $y_1$ | Position | V, F, P | V, F | $\mathcal{U}(3,4)$ | $\mathcal{U}_{\log}(10^0,10^1)$ |
| | | $z_2$ | Position | V, F, P | V, F | $\mathcal{U}(1,2)$ | $\mathcal{U}_{\log}(10^0,10^1)$ |
| | | $z_1$ | Position | V, F, P | V, F | $\mathcal{U}(3,4)$ | $\mathcal{U}_{\log}(10^0,10^1)$ |
| I.15.3t | $t_1 = \dfrac{t-ux/c^2}{\sqrt{1-u^2/c^2}}$ | $t_1$ | Time | V, F | V, F | N/A | N/A |
| | | $t$ | Time | V, F, P | V, F, NN | $\mathcal{U}(1,5)$ | $\mathcal{U}_{\log}(10^{-6},10^{-4})$ |
| | | $u$ | Velocity | V, F, P | V, F | $\mathcal{U}(1,2)$ | $\mathcal{U}_{\log}(10^5,10^7)$ |
| | | $x$ | Position | V, F, P | V, F | $\mathcal{U}(1,5)$ | $\mathcal{U}_{\log}(10^0,10^2)$ |
| | | $c$ | Speed of light | V, F, P | C, F, P | $\mathcal{U}(3,10)$ | $2.998\times10^8$ |
| I.15.3x | $x_1 = \dfrac{x-ut}{\sqrt{1-u^2/c^2}}$ | $x_1$ | Position | V, F, P | V, F | N/A | N/A |
| | | $x$ | Position | V, F, P | V, F | $\mathcal{U}(5,10)$ | $\mathcal{U}_{\log}(10^0,10^2)$ |
| | | $u$ | Velocity | V, F, P | V, F | $\mathcal{U}(1,2)$ | $\mathcal{U}_{\log}(10^6,10^8)$ |
| | | $t$ | Time | V, F, P | V, F, P | $\mathcal{U}(1,2)$ | $\mathcal{U}_{\log}(10^{-6},10^{-4})$ |
| | | $c$ | Speed of light | V, F, P | C, F, P | $\mathcal{U}(3,20)$ | $2.998\times10^8$ |
| I.29.16 | $x = \sqrt{x_1^2+x_2^2+2x_1 x_2 \cos(\theta_1-\theta_2)}$ | $x$ | Wavelength | V, F, P | V, F, P | N/A | N/A |
| | | $x_1$ | Wavelength | V, F, P | V, F, P | $\mathcal{U}(1,5)$ | $\mathcal{U}_{\log}(10^{-1},10^1)$ |
| | | $x_2$ | Wavelength | V, F, P | V, F, P | $\mathcal{U}(1,5)$ | $\mathcal{U}_{\log}(10^{-1},10^1)$ |
| | | $\theta_1$ | Angle | V, F, P | V, F, NN | $\mathcal{U}(1,5)$ | $\mathcal{U}(0,2\pi)$ |
| | | $\theta_2$ | Angle | V, F, P | V, F, NN | $\mathcal{U}(1,5)$ | $\mathcal{U}(0,2\pi)$ |
| I.30.3 | $I = I_0 \dfrac{\sin^2(n\theta/2)}{\sin^2(\theta/2)}$ | $I$ | Amplitude of combined wave | V, F, P | V, F, P | N/A | N/A |
| | | $I_0$ | Amplitude of wave | V, F, P | V, F, P | $\mathcal{U}(1,5)$ | $\mathcal{U}_{\log}(10^1,10^3)$ |
| | | $n$ | Number of waves | V, F, P | V, I, P | $\mathcal{U}(1,5)$ | $\mathcal{U}_{\log}(10^1,10^3)$ |
| | | $\theta$ | Phase difference | V, F, P | V, F | $\mathcal{U}(1,5)$ | $\mathcal{U}(-2\pi,2\pi)$ |
| I.32.17 | $P = \left(\frac{1}{2}\epsilon c E^2\right)\left(\frac{8\pi r^2}{3}\right)\left(\frac{\omega^4}{(\omega^2-\omega_0^2)^2}\right)$ | $P$ | Energy | V, F, P | V, F, P | N/A | N/A |
| | | $\epsilon$ | Vacuum permittivity | V, F, P | C, F, P | $\mathcal{U}(1,2)$ | $8.854\times10^{-12}$ |
| | | $c$ | Speed of light | V, F, P | C, F, P | $\mathcal{U}(1,2)$ | $2.998\times10^8$ |
| | | $E$ | Magnitude of electric field | V, F, P | V, F | $\mathcal{U}(1,2)$ | $\mathcal{U}_{\log}(10^1,10^3)$ |
| | | $r$ | Radius | V, F, P | V, F, P | $\mathcal{U}(1,2)$ | $\mathcal{U}_{\log}(10^{-2},10^0)$ |
| | | $\omega$ | Frequency of electromagnetic waves | V, F, P | V, F | $\mathcal{U}(1,2)$ | $\mathcal{U}_{\log}(10^9,10^{11})$ |
| | | $\omega_0$ | Frequency of electromagnetic waves | V, F, P | V, F | $\mathcal{U}(3,5)$ | $\mathcal{U}_{\log}(10^9,10^{11})$ |
| I.34.14 | $\omega = \dfrac{1+v/c}{\sqrt{1-v^2/c^2}}\omega_0$ | $\omega$ | Frequency of electromagnetic waves | V, F, P | V, F, P | N/A | N/A |
| | | $v$ | Velocity | V, F, P | V, F | $\mathcal{U}(1,2)$ | $\mathcal{U}_{\log}(10^6,10^8)$ |
| | | $c$ | Speed of light | V, F, P | C, F, P | $\mathcal{U}(3,10)$ | $2.998\times10^8$ |
| | | $\omega_0$ | Frequency of electromagnetic waves | V, F, P | V, F, P | $\mathcal{U}(1,5)$ | $\mathcal{U}_{\log}(10^9,10^{11})$ |

Table 15: Hard set of our proposed datasets (part 2).

| Eq. ID | Formula | Symbols | | Properties | | Distributions | |
|---|---|---|---|---|---|---|---|
| | | | | Original | Ours | Original | Ours |
| I.37.4 | $I_{12} = I_1 + I_2$ $+ 2\sqrt{I_1 I_2}\cos\delta$ | $I_{12}$ | Amplitude of wave | V, F, NN | V, F, NN | N/A | N/A |
| | | $I_1$ | Amplitude of wave | V, F, P | V, F, P | $\mathcal{U}(1,5)$ | $\mathcal{U}_{\log}(10^{-3},10^{-1})$ |
| | | $I_2$ | Amplitude of wave | V, F, P | V, F, P | $\mathcal{U}(1,5)$ | $\mathcal{U}_{\log}(10^{-3},10^{-1})$ |
| | | $\delta$ | Phase difference | V, F, P | V, F, NN | $\mathcal{U}(1,5)$ | $\mathcal{U}(0,2\pi)$ |
| I.39.22 | $P = \frac{nkT}{V}$ | $P$ | Pressure | V, F, P | V, F, P | N/A | N/A |
| | | $n$ | Number of molecules | V, F, P | V, I⋆, P | $\mathcal{U}(1,5)$ | $\mathcal{U}_{\log}(10^{23},10^{25})$ |
| | | $k$ | Boltzmann constant | V, F, P | C, F, P | $\mathcal{U}(1,5)$ | $1.381 \times 10^{-23}$ |
| | | $T$ | Temperature | V, F, P | V, F, P | $\mathcal{U}(1,5)$ | $\mathcal{U}_{\log}(10^1,10^3)$ |
| | | $V$ | Volume | V, F, P | V, F, P | $\mathcal{U}(1,5)$ | $\mathcal{U}_{\log}(10^{-5},10^{-3})$ |
| I.40.1 | $n = n_0 \exp\left(-mgx/kT\right)$ | $n$ | Molecular density | V, F, P | V, F, P | N/A | N/A |
| | | $n_0$ | Molecular density | V, F, P | V, F, P | $\mathcal{U}(1,5)$ | $\mathcal{U}_{\log}(10^{25},10^{27})$ |
| | | $m$ | Mass | V, F, P | V, F, P | $\mathcal{U}(1,5)$ | $\mathcal{U}_{\log}(10^{-24},10^{-22})$ |
| | | $g$ | Gravitational acceleration | V, F, P | C, F, P | $\mathcal{U}(1,5)$ | $9.807 \times 10^0$ |
| | | $x$ | Height | V, F, P | V, F | $\mathcal{U}(1,5)$ | $\mathcal{U}_{\log}(10^{-2},10^0)$ |
| | | $k$ | Boltzmann constant | V, F, P | C, F, P | $\mathcal{U}(1,5)$ | $1.381 \times 10^{-23}$ |
| | | $T$ | Temperature | V, F, P | V, F, P | $\mathcal{U}(1,5)$ | $\mathcal{U}_{\log}(10^1,10^3)$ |
| I.41.16 | $L_{\text{rad}} = \frac{h}{2\pi}$ $\frac{\omega^3}{\pi^2 c^2 (\exp(h\omega/2\pi kT)-1)}$ | $L_{\text{rad}}$ | Radiation per frequency | V, F, P | V, F, P | N/A | N/A |
| | | $h$ | Planck constant | V, F, P | C, F, P | $\mathcal{U}(1,5)$ | $6.626 \times 10^{-34}$ |
| | | $\omega$ | Frequency of electromagnetic wave | V, F, P | V, F, P | $\mathcal{U}(1,5)$ | $\mathcal{U}_{\log}(10^{-1},10^1)$ |
| | | $c$ | Speed of light | V, F, P | C, F, P | $\mathcal{U}(1,5)$ | $2.998 \times 10^8$ |
| | | $k$ | Boltzmann constant | V, F, P | C, F, P | $\mathcal{U}(1,5)$ | $1.381 \times 10^{-23}$ |
| | | $T$ | Temperature | V, F, P | V, F, P | $\mathcal{U}(1,5)$ | $\mathcal{U}_{\log}(10^1,10^3)$ |
| I.44.4 | $Q = nkT \ln(\frac{V_2}{V_1})$ | $Q$ | Energy | V, F | V, F | N/A | N/A |
| | | $n$ | Number of molecules | V, F, P | V, I⋆, P | $\mathcal{U}(1,5)$ | $\mathcal{U}_{\log}(10^{24},10^{26})$ |
| | | $k$ | Boltzmann constant | V, F, P | C, F, P | $\mathcal{U}(1,5)$ | $1.381 \times 10^{-23}$ |
| | | $T$ | Temperature | V, F, P | V, F, P | $\mathcal{U}(1,5)$ | $\mathcal{U}_{\log}(10^1,10^3)$ |
| | | $V_2$ | Volume | V, F, P | V, F, P | $\mathcal{U}(1,5)$ | $\mathcal{U}_{\log}(10^{-5},10^{-3})$ |
| | | $V_1$ | Volume | V, F, P | V, F, P | $\mathcal{U}(1,5)$ | $\mathcal{U}_{\log}(10^{-5},10^{-3})$ |
| I.50.26 | $x = K\left(\cos\omega t + \epsilon \cos^2 \omega t\right)$ | $x$ | Amplitude | V, F | V, F | N/A | N/A |
| | | $K$ | Amplitude | V, F, P | V, F, P | $\mathcal{U}(1,3)$ | $\mathcal{U}_{\log}(10^{-1},10^1)$ |
| | | $\omega$ | Angular velocity | V, F, P | V, F | $\mathcal{U}(1,3)$ | $\mathcal{U}_{\log}(10^1,10^3)$ |
| | | $t$ | Time | V, F, P | V, F, NN | $\mathcal{U}(1,3)$ | $\mathcal{U}_{\log}(10^{-3},10^{-1})$ |
| | | $\epsilon$ | Variable | V, F, P | V, F | $\mathcal{U}(1,3)$ | $\mathcal{U}_{\log}(10^{-3},10^{-1})$ |
| II.6.15a | $E = \frac{p}{4\pi\epsilon}\frac{3z}{r^5}\sqrt{x^2+y^2}$ | $E$ | Electric field | V, F, P | V, F | N/A | N/A |
| | | $p$ | Electric dipole moment | V, F, P | V, F | $\mathcal{U}(1,3)$ | $\mathcal{U}_{\log}(10^{-22},10^{-20})$ |
| | | $\epsilon$ | Vacuum permittivity | V, F, P | C, F, P | $\mathcal{U}(1,3)$ | $8.854 \times 10^{-12}$ |
| | | $z$ | Position | V, F, P | V, F | $\mathcal{U}(1,3)$ | $\mathcal{U}_{\log}(10^{-10},10^{-8})$ |
| | | $r$ | Distance | V, F, P | V, F, P | $\mathcal{U}(1,3)$ | $\mathcal{U}_{\log}(10^{-10},10^{-8})$ |
| | | $x$ | Position | V, F, P | V, F | $\mathcal{U}(1,3)$ | $\mathcal{U}_{\log}(10^{-10},10^{-8})$ |
| | | $y$ | Position | V, F, P | V, F | $\mathcal{U}(1,3)$ | $\mathcal{U}_{\log}(10^{-10},10^{-8})$ |
| II.6.15b | $E = \frac{p}{4\pi\epsilon}\frac{3\cos\theta\sin\theta}{r^3}$ | $E$ | Electric field | V, F | V, F | N/A | N/A |
| | | $p$ | Electric dipole moment | V, F, P | V, F | $\mathcal{U}(1,3)$ | $\mathcal{U}_{\log}(10^{-22},10^{-20})$ |
| | | $\epsilon$ | Vacuum permittivity | V, F, P | C, F, P | $\mathcal{U}(1,3)$ | $8.854 \times 10^{-12}$ |
| | | $\theta$ | Angle | V, F, P | V, F | $\mathcal{U}(1,3)$ | $\mathcal{U}(0,\pi)$ |
| | | $r$ | Distance | V, F, P | V, F, P | $\mathcal{U}(1,3)$ | $\mathcal{U}_{\log}(10^{-10},10^{-8})$ |
| II.11.17 | $n = n_0 \left(1 + \frac{p_0 E \cos\theta}{kT}\right)$ | $n$ | Number of polar molecules per angle per unit volume | V, F | V, F | N/A | N/A |
| | | $n_0$ | Number of molecules per unit volume | V, F, P | V, F, P | $\mathcal{U}(1,3)$ | $\mathcal{U}_{\log}(10^{27},10^{29})$ |
| | | $p_0$ | Electric dipole moment | V, F, P | V, F | $\mathcal{U}(1,3)$ | $\mathcal{U}_{\log}(10^{-22},10^{-20})$ |
| | | $E$ | Magnitude of electric field | V, F, P | V, F | $\mathcal{U}(1,3)$ | $\mathcal{U}_{\log}(10^1,10^3)$ |
| | | $\theta$ | Angle | V, F, P | V, F, NN | $\mathcal{U}(1,3)$ | $\mathcal{U}(0,2\pi)$ |
| | | $k$ | Boltzmann constant | V, F, P | C, F, P | $\mathcal{U}(1,3)$ | $1.381 \times 10^{-23}$ |
| | | $T$ | Temperature | V, F, P | V, F, P | $\mathcal{U}(1,3)$ | $\mathcal{U}_{\log}(10^1,10^3)$ |
| II.11.20 | $P = \frac{n_0 p_0^2 E}{3kT}$ | $P$ | Polarizability | V, F, P | V, F | N/A | N/A |
| | | $n_0$ | Number of atom | V, F, P | V, I⋆, P | $\mathcal{U}(1,5)$ | $\mathcal{U}_{\log}(10^{23},10^{25})$ |
| | | $p_0$ | Electric dipole moment | V, F, P | V, F | $\mathcal{U}(1,5)$ | $\mathcal{U}_{\log}(10^{-22},10^{-20})$ |
| | | $E$ | Magnitude of electric field | V, F, P | V, F | $\mathcal{U}(1,5)$ | $\mathcal{U}_{\log}(10^1,10^3)$ |
| | | $k$ | Boltzmann constant | V, F, P | C, F, P | $\mathcal{U}(1,5)$ | $1.381 \times 10^{-23}$ |
| | | $T$ | Temperature | V, F, P | V, F, P | $\mathcal{U}(1,5)$ | $\mathcal{U}_{\log}(10^1,10^3)$ |

Table 16: Hard set of our proposed datasets (part 3).

| Eq. ID | Formula | Symbols | | Properties | | Distributions | |
|---|---|---|---|---|---|---|---|
| | | | | Original | Ours | Original | Ours |
| II.11.27 | $P = \frac{N\alpha}{1-(N\alpha/3)}\epsilon E$ | $P$ | Polarizability | V, F, P | V, F | N/A | N/A |
| | | $N$ | Number of atom | V, F, NN | V, I$\star$, P | $\mathcal{U}(0,1)$ | $\mathcal{U}_{\log}(10^{23}, 10^{25})$ |
| | | $\alpha$ | Molecular polarizability | V, F, NN | V, F, P | $\mathcal{U}(0,1)$ | $\mathcal{U}_{\log}(10^{-33}, 10^{-31})$ |
| | | $\epsilon$ | Vacuum permittivity | V, F, P | C, F, P | $\mathcal{U}(1,2)$ | $8.854 \times 10^{-12}$ |
| | | $E$ | Magnitude of electric field | V, F, P | V, F, P | $\mathcal{U}(1,2)$ | $\mathcal{U}_{\log}(10^1, 10^3)$ |
| II.11.28 | $\kappa = 1 + \frac{N\alpha}{1-(N\alpha/3)}$ | $\kappa$ | Electric dipole moment per unit volume | V, F, P | V, F | N/A | N/A |
| | | $N$ | Number of electric dipoles | V, F, NN | V, I$\star$, P | $\mathcal{U}(0,1)$ | $\mathcal{U}_{\log}(10^{23}, 10^{25})$ |
| | | $\alpha$ | Molecular polarizability | V, F, NN | V, F, P | $\mathcal{U}(0,1)$ | $\mathcal{U}_{\log}(10^{-33}, 10^{-31})$ |
| II.13.23 | $\rho = \frac{\rho_0}{\sqrt{1-v^2/c^2}}$ | $\rho$ | Electric charge density | V, F, P | V, F, P | N/A | N/A |
| | | $\rho_0$ | Electric charge density | V, F, P | V, F, P | $\mathcal{U}(1,5)$ | $\mathcal{U}_{\log}(10^{27}, 10^{29})$ |
| | | $v$ | Velocity | V, F, P | V, F, P | $\mathcal{U}(1,2)$ | $\mathcal{U}_{\log}(10^6, 10^8)$ |
| | | $c$ | Speed of light | V, F, P | C, F, P | $\mathcal{U}(3,10)$ | $2.998 \times 10^8$ |
| II.13.34 | $j = \frac{\rho_0 v}{\sqrt{1-v^2/c^2}}$ | $j$ | Electric current | V, F, P | V, F, P | N/A | N/A |
| | | $\rho_0$ | Electric charge density | V, F, P | V, F, P | $\mathcal{U}(1,5)$ | $\mathcal{U}_{\log}(10^{27}, 10^{29})$ |
| | | $v$ | Velocity | V, F, P | V, F, P | $\mathcal{U}(1,2)$ | $\mathcal{U}_{\log}(10^6, 10^8)$ |
| | | $c$ | Speed of light | V, F, P | C, F, P | $\mathcal{U}(3,10)$ | $2.998 \times 10^8$ |
| II.24.17 | $k = \sqrt{\omega^2/c^2 - \pi^2/a^2}$ | $k$ | Wavenumber | V, F, P | V, F, P | N/A | N/A |
| | | $\omega$ | Angular velocity | V, F, P | V, F | $\mathcal{U}(4,6)$ | $\mathcal{U}_{\log}(10^9, 10^{11})$ |
| | | $c$ | Speed of light | V, F, P | C, F, P | $\mathcal{U}(1,2)$ | $2.998 \times 10^8$ |
| | | $a$ | Length | V, F, P | V, F, P | $\mathcal{U}(2,4)$ | $\mathcal{U}_{\log}(10^{-3}, 10^{-1})$ |
| II.35.18 | $a = \frac{N}{\exp(\mu B/kT) + \exp(-\mu B/kT)}$ | $a$ | Number of atoms with the equivalent magnetic moment | V, F, P | V, I$\star$, P | N/A | N/A |
| | | $N$ | Number of atoms per unit volume | V, F, P | V, I$\star$, P | $\mathcal{U}(1,3)$ | $\mathcal{U}_{\log}(10^{23}, 10^{25})$ |
| | | $\mu$ | Magnetic moment | V, F, P | V, F, P | $\mathcal{U}(1,3)$ | $\mathcal{U}_{\log}(10^{-25}, 10^{-23})$ |
| | | $B$ | Magnetic flux density | V, F, P | V, F, P | $\mathcal{U}(1,3)$ | $\mathcal{U}_{\log}(10^{-3}, 10^{-1})$ |
| | | $k$ | Boltzmann constant | V, F, P | C, F, P | $\mathcal{U}(1,3)$ | $1.381 \times 10^{-23}$ |
| | | $T$ | Temperature | V, F, P | V, F, P | $\mathcal{U}(1,3)$ | $\mathcal{U}_{\log}(10^1, 10^3)$ |
| II.35.21 | $M = N\mu \tanh\left(\frac{\mu B}{kT}\right)$ | $M$ | Number of magnetized atoms | V, F, P | V, I$\star$, P | N/A | N/A |
| | | $N$ | Number of atom | V, F, P | V, I$\star$, P | $\mathcal{U}(1,5)$ | $\mathcal{U}_{\log}(10^{23}, 10^{25})$ |
| | | $\mu$ | Magnetic moment | V, F, P | V, F, P | $\mathcal{U}(1,5)$ | $\mathcal{U}_{\log}(10^{-25}, 10^{-23})$ |
| | | $B$ | Magnetic flux density | V, F, P | V, F, P | $\mathcal{U}(1,5)$ | $\mathcal{U}_{\log}(10^{-3}, 10^{-1})$ |
| | | $k$ | Boltzmann constant | V, F, P | C, F, P | $\mathcal{U}(1,5)$ | $1.381 \times 10^{-23}$ |
| | | $T$ | Temperature | V, F, P | V, F, P | $\mathcal{U}(1,5)$ | $\mathcal{U}_{\log}(10^1, 10^3)$ |
| II.36.38 | $x = \frac{\mu H}{kT} + \frac{\mu\lambda}{\epsilon c^2 kT}M$ | $x$ | Parameter of magnetization | V, F, P | V, F | N/A | N/A |
| | | $\mu$ | Magnetic moment | V, F, P | V, F | $\mathcal{U}(1,3)$ | $\mathcal{U}_{\log}(10^{-25}, 10^{-23})$ |
| | | $H$ | Magnetic field strength | V, F, P | V, F | $\mathcal{U}(1,3)$ | $\mathcal{U}_{\log}(10^{-3}, 10^{-1})$ |
| | | $k$ | Boltzmann constant | V, F, P | C, F, P | $\mathcal{U}(1,3)$ | $1.381 \times 10^{-23}$ |
| | | $T$ | Temperature | V, F, P | V, F, P | $\mathcal{U}(1,3)$ | $\mathcal{U}_{\log}(10^1, 10^3)$ |
| | | $\lambda$ | Constant | V, F, P | V, F, NN | $\mathcal{U}(1,3)$ | $\mathcal{U}(0,1)$ |
| | | $\epsilon$ | Vacuum permittivity | V, F, P | C, F, P | $\mathcal{U}(1,3)$ | $8.854 \times 10^{-12}$ |
| | | $c$ | Speed of light | V, F, P | C, F, P | $\mathcal{U}(1,3)$ | $2.998 \times 10^8$ |
| | | $M$ | Number of magnetized atoms | V, F, P | V, I$\star$, P | $\mathcal{U}(1,3)$ | $\mathcal{U}_{\log}(10^{23}, 10^{25})$ |
| III.4.33 | $E = \frac{h\omega}{2\pi(\exp(h\omega/2\pi kT)-1)}$ | $E$ | Energy | V, F, P | V, F, P | N/A | N/A |
| | | $h$ | Planck constant | V, F, P | C, F, P | $\mathcal{U}(1,5)$ | $6.626 \times 10^{-34}$ |
| | | $\omega$ | Frequency | V, F, P | V, F, P | $\mathcal{U}(1,5)$ | $\mathcal{U}_{\log}(10^8, 10^{10})$ |
| | | $k$ | Boltzmann constant | V, F, P | C, F, P | $\mathcal{U}(1,5)$ | $1.381 \times 10^{-23}$ |
| | | $T$ | Temperature | V, F, P | V, F, P | $\mathcal{U}(1,5)$ | $\mathcal{U}_{\log}(10^1, 10^3)$ |
| III.9.52 | $P_{\mathrm{I}\to\mathrm{II}} = \left(\frac{2\pi\mu Et}{h}\right)^2 \frac{\sin^2\left((\omega-\omega_0)\,t/2\right)}{((\omega-\omega_0)\,t/2)^2}$ | $P_{\mathrm{I}\to\mathrm{II}}$ | Probability | V, F, P | V, F, NN | N/A | N/A |
| | | $\mu$ | Electric dipole moment | V, F, P | V, F | $\mathcal{U}(1,3)$ | $\mathcal{U}_{\log}(10^{-22}, 10^{-20})$ |
| | | $E$ | Magnitude of electric field | V, F, P | V, F | $\mathcal{U}(1,3)$ | $\mathcal{U}_{\log}(10^1, 10^3)$ |
| | | $t$ | Time | V, F, P | V, F, NN | $\mathcal{U}(1,3)$ | $\mathcal{U}_{\log}(10^{-18}, 10^{-16})$ |
| | | $h$ | Planck constant | V, F, P | C, F, P | $\mathcal{U}(1,3)$ | $6.626 \times 10^{-34}$ |
| | | $\omega$ | Frequency | V, F, P | V, F, P | $\mathcal{U}(1,5)$ | $\mathcal{U}_{\log}(10^8, 10^{10})$ |
| | | $\omega_0$ | Resonant frequency | V, F, P | V, F, P | $\mathcal{U}(1,5)$ | $\mathcal{U}_{\log}(10^8, 10^{10})$ |

Table 17: Hard set of our proposed datasets (part 4).

| Eq. ID | Formula | Symbols | | Properties Original | Properties Ours | Distributions Original | Distributions Ours |
|---|---|---|---|---|---|---|---|
| III.10.19 | $E = \mu\sqrt{B_x^2 + B_y^2 + B_z^2}$ | $E$ | Energy | V, F, P | V, F | N/A | N/A |
| | | $\mu$ | Magnetic moment | V, F, P | V, F | $\mathcal{U}(1,5)$ | $\mathcal{U}_{\log}(10^{-25}, 10^{-23})$ |
| | | $B_x$ | Element of magnetic field | V, F, P | V, F | $\mathcal{U}(1,5)$ | $\mathcal{U}_{\log}(10^{-3}, 10^{-1})$ |
| | | $B_y$ | Element of magnetic field | V, F, P | V, F | $\mathcal{U}(1,5)$ | $\mathcal{U}_{\log}(10^{-3}, 10^{-1})$ |
| | | $B_z$ | Element of magnetic field | V, F, P | V, F | $\mathcal{U}(1,5)$ | $\mathcal{U}_{\log}(10^{-3}, 10^{-1})$ |
| III.21.20 | $J = -\rho\frac{q}{m}A$ | $J$ | Electric current | V, F, N | V, F | N/A | N/A |
| | | $\rho$ | Electric charge density | V, F, P | V, F, N | $\mathcal{U}(1,5)$ | $\mathcal{U}_{\log}(10^{27}, 10^{29})$ |
| | | $q$ | Electric charge | V, F, P | V, F, N | $\mathcal{U}(1,5)$ | $\mathcal{U}_{\log}(10^{-11}, 10^{-9})$ |
| | | $A$ | Magnetic vector potential | V, F, P | V, F | $\mathcal{U}(1,5)$ | $\mathcal{U}_{\log}(10^{-3}, 10^{-1})$ |
| | | $m$ | Mass | V, F, P | V, F, P | $\mathcal{U}(1,5)$ | $\mathcal{U}_{\log}(10^{-30}, 10^{-28})$ |
| B1 | $A = \left(\frac{Z_1 Z_2 \alpha h c}{4E\sin^2(\theta/2)}\right)^2$ | $A$ | Differential scattering cross section | V, F, P | V, F, P | N/A | N/A |
| | | $Z_1$ | Atomic number | V, F, P | V, I, P | $\mathcal{U}(1,2)$ | $\mathcal{U}_{\log}(10^0, 10^1)$ |
| | | $Z_2$ | Atomic number | V, F, P | V, I, P | $\mathcal{U}(1,2)$ | $\mathcal{U}_{\log}(10^0, 10^1)$ |
| | | $\alpha$ | Fine structure constant | V, F, P | C, F, P | $\mathcal{U}(1,5)$ | $7.297 \times 10^{-3}$ |
| | | $h$ | Dirac's constant | V, F, P | C, F, P | $\mathcal{U}(1,2)$ | $1.055 \times 10^{-34}$ |
| | | $c$ | Speed of light | V, F, P | C, F, P | $\mathcal{U}(1,2)$ | $2.998 \times 10^8$ |
| | | $E$ | Non-relativistic kinetic energy | V, F, P | V, F, P | $\mathcal{U}(1,3)$ | $\mathcal{U}_{\log}(10^{-18}, 10^{-16})$ |
| | | $\theta$ | Scattering angle | V, F, P | V, F, NN | $\mathcal{U}(1,3)$ | $\mathcal{U}(0, 2\pi)$ |
| B2 | $k = \frac{mk_G}{L^2}$ $\left(1 + \sqrt{1 + \frac{2EL^2}{mk_G^2}}\cos(\theta_1 - \theta_2)\right)$ | $k$ | Inverse radius | V, F | V, F | N/A | N/A |
| | | $m$ | Mass (The Earth) | V, F, P | V, F, P | $\mathcal{U}(1,3)$ | $\mathcal{U}_{\log}(10^{23}, 10^{25})$ |
| | | $k_G$ | Variable | V, F, P | V, F, P | $\mathcal{U}(1,3)$ | $\mathcal{U}_{\log}(10^9, 10^{11})$ |
| | | $L$ | Angular Momentum | V, F, P | V, F, P | $\mathcal{U}(1,3)$ | $\mathcal{U}_{\log}(10^8, 10^{10})$ |
| | | $E$ | Energy | V, F, P | V, F, P | $\mathcal{U}(1,3)$ | $\mathcal{U}_{\log}(10^{25}, 10^{27})$ |
| | | $\theta_1$ | Angle | V, F, P | V, F, NN | $\mathcal{U}(0,6)$ | $\mathcal{U}(0, 2\pi)$ |
| | | $\theta_2$ | Angle | V, F, P | V, F, NN | $\mathcal{U}(0,6)$ | $\mathcal{U}(0, 2\pi)$ |
| B3 | $r = \frac{d(1-\alpha^2)}{1+\alpha\cos(\theta_1-\theta_2)}$ | $r$ | Distance | V, F, N | V, F, P | N/A | N/A |
| | | $d$ | Semi-major axis of elliptical orbit | V, F, P | V, F, P | $\mathcal{U}(1,3)$ | $\mathcal{U}_{\log}(10^8, 10^{10})$ |
| | | $\alpha$ | Orbital eccentricity | V, F, P | V, F, P | $\mathcal{U}(2,4)$ | $\mathcal{U}(0,1)$ |
| | | $\theta_1$ | Angle | V, F, P | V, F, NN | $\mathcal{U}(4,5)$ | $\mathcal{U}(0, 2\pi)$ |
| | | $\theta_2$ | Angle | V, F, P | V, F, NN | $\mathcal{U}(4,5)$ | $\mathcal{U}(0, 2\pi)$ |
| B4 | $v = \sqrt{\frac{2}{m}\left(E - U - \frac{L^2}{2mr^2}\right)}$ | $v$ | Velocity | V, F, P | V, F, P | N/A | N/A |
| | | $m$ | Mass (The Earth) | V, F, P | V, F, P | $\mathcal{U}(1,3)$ | $\mathcal{U}_{\log}(10^{23}, 10^{25})$ |
| | | $E$ | Energy | V, F, P | V, F, P | $\mathcal{U}(8,12)$ | $\mathcal{U}_{\log}(10^{25}, 10^{27})$ |
| | | $U$ | Potential energy | V, F, P | V, F, P | $\mathcal{U}(1,3)$ | $\mathcal{U}_{\log}(10^{25}, 10^{27})$ |
| | | $L$ | Angular momentum | V, F, P | V, F | $\mathcal{U}(1,3)$ | $\mathcal{U}_{\log}(10^8, 10^{10})$ |
| | | $r$ | Distance | V, F, P | V, F, P | $\mathcal{U}(1,3)$ | $\mathcal{U}_{\log}(10^8, 10^{10})$ |
| B5 | $t = \frac{2\pi d^{3/2}}{\sqrt{G(m_1+m_2)}}$ | $t$ | Orbital period | V, F, P | V, F, P | N/A | N/A |
| | | $d$ | Semimajor axis of elliptical orbit | V, F, P | V, F, P | $\mathcal{U}(1,3)$ | $\mathcal{U}_{\log}(10^8, 10^{10})$ |
| | | $G$ | Gravitational constant | V, F, P | C, F, P | $\mathcal{U}(1,3)$ | $6.674 \times 10^{-11}$ |
| | | $m_1$ | Mass (The Earth) | V, F, P | V, F, P | $\mathcal{U}(1,3)$ | $\mathcal{U}_{\log}(10^{23}, 10^{25})$ |
| | | $m_2$ | Mass (The Earth) | V, F, P | V, F, P | $\mathcal{U}(1,3)$ | $\mathcal{U}_{\log}(10^{23}, 10^{25})$ |
| B6 | $\alpha = \sqrt{1 + \frac{2\epsilon^2 E L^2}{m(Z_1 Z_2 q^2)^2}}$ | $\alpha$ | Orbital eccentricity | V, F, P | V, F, P | N/A | N/A |
| | | $\epsilon$ | Energy | V, F, P | V, F | $\mathcal{U}(1,3)$ | $\mathcal{U}_{\log}(10^{-18}, 10^{-16})$ |
| | | $E$ | Energy | V, F, P | V, F, P | $\mathcal{U}(1,3)$ | $\mathcal{U}_{\log}(10^{-18}, 10^{-16})$ |
| | | $L$ | Distance | V, F, P | V, F, P | $\mathcal{U}(1,3)$ | $\mathcal{U}_{\log}(10^{-10}, 10^{-8})$ |
| | | $m$ | Mass | V, F, P | V, F, P | $\mathcal{U}(1,3)$ | $\mathcal{U}_{\log}(10^{-30}, 10^{-28})$ |
| | | $Z_1$ | Atomic number | V, F, P | V, I, P | $\mathcal{U}(1,3)$ | $\mathcal{U}_{\log}(10^0, 10^1)$ |
| | | $Z_2$ | Atomic number | V, F, P | V, I, P | $\mathcal{U}(1,3)$ | $\mathcal{U}_{\log}(10^0, 10^1)$ |
| | | $q$ | Electric charge | V, F, P | V, F | $\mathcal{U}(1,3)$ | $\mathcal{U}_{\log}(10^{-11}, 10^{-9})$ |
| B7 | $H = \sqrt{\frac{8\pi G\rho}{3} - \frac{k_f c^2}{a_f^2}}$ | $H$ | Hubble's constant | V, F, P | V, F, P | N/A | N/A |
| | | $G$ | Gravitational constant | V, F, P | C, F, P | $\mathcal{U}(1,3)$ | $6.674 \times 10^{-11}$ |
| | | $\rho$ | Density of the Universe | V, F, P | V, F, P | $\mathcal{U}(1,3)$ | $\mathcal{U}_{\log}(10^{-28}, 10^{-26})$ |
| | | $k_f$ | Spacetime curvature | V, F, P | V, I | $\mathcal{U}(1,2)$ | $\mathcal{U}(-1,1)$ |
| | | $c$ | Speed of light | V, F, P | C, F, P | $\mathcal{U}(1,2)$ | $2.998 \times 10^8$ |
| | | $a_f$ | Radius | V, F, P | V, F, P | $\mathcal{U}(1,3)$ | $\mathcal{U}_{\log}(10^{22}, 10^{24})$ |
| B9 | $P = $ $-\frac{32}{5}\frac{G^4}{c^5}\frac{(m_1 m_2)^2(m_1+m_2)}{r^5}$ | $P$ | Gravitational wave energy | V, F, N | V, F, N | N/A | N/A |
| | | $G$ | Gravitational constant | V, F, P | C, F, P | $\mathcal{U}(1,2)$ | $6.674 \times 10^{-11}$ |
| | | $c$ | Speed of light | V, F, P | C, F, P | $\mathcal{U}(1,2)$ | $2.998 \times 10^8$ |
| | | $m_1$ | Mass | V, F, P | V, F, P | $\mathcal{U}(1,5)$ | $\mathcal{U}_{\log}(10^{23}, 10^{25})$ |
| | | $m_2$ | Mass | V, F, P | V, F, P | $\mathcal{U}(1,5)$ | $\mathcal{U}_{\log}(10^{23}, 10^{25})$ |
| | | $r$ | Distance | V, F, P | V, F, P | $\mathcal{U}(1,2)$ | $\mathcal{U}_{\log}(10^8, 10^{10})$ |

Table 18: Hard set of our proposed datasets (part 5).

| Eq. ID | Formula | Symbols | | Properties | | Distributions | |
|---|---|---|---|---|---|---|---|
| | | | | Original | Ours | Original | Ours |
| B10 | $\cos\theta_1 = \frac{\cos\theta_2 - v/c}{(1-v/c)\cos\theta_2}$ | $\cos\theta_1$ | Value | V, F | V, F | N/A | N/A |
| | | $\theta_2$ | Angle | V, F, P | V, F, NN | $\mathcal{U}(1,3)$ | $\mathcal{U}(0,\pi)$ |
| | | $v$ | Velocity | V, F, P | V, F | $\mathcal{U}(1,3)$ | $\mathcal{U}_{\log}(10^6,10^8)$ |
| | | $c$ | Speed of light | V, F, P | C, F, P | $\mathcal{U}(4,6)$ | $2.998\times10^8$ |
| B11 | $I = I_0\left(\frac{\sin(\alpha/2)}{\alpha/2}\frac{\sin(N\delta/2)}{\sin(\delta/2)}\right)^2$ | $I$ | Wave intensity | V, F, P | V, F, P | N/A | N/A |
| | | $I_0$ | Amplitude of wave | V, F, P | V, F, P | $\mathcal{U}(1,3)$ | $\mathcal{U}_{\log}(10^{-3},10^{-1})$ |
| | | $\alpha$ | Wavelength of X-ray | V, F, P | V, F, P | $\mathcal{U}(1,3)$ | $\mathcal{U}_{\log}(10^{-11},10^{-9})$ |
| | | $N$ | Number of phase difference | V, F, P | V, I, P | $\mathcal{U}(1,2)$ | $\mathcal{U}_{\log}(10^0,10^2)$ |
| | | $\delta$ | Wavelength of X-ray | V, F, P | V, F, P | $\mathcal{U}(1,3)$ | $\mathcal{U}_{\log}(10^{-11},10^{-9})$ |
| B12 | $F = \frac{q}{4\pi\epsilon y^2}$ $\left(4\pi\epsilon V_e d - \frac{qdy^3}{(y^2-d^2)^2}\right)$ | $F$ | Force | V, F, P | V, F | N/A | N/A |
| | | $q$ | Electric charge | V, F, P | V, F | $\mathcal{U}(1,5)$ | $\mathcal{U}_{\log}(10^{-3},10^{-1})$ |
| | | $\epsilon$ | Permittivity | V, F, P | V, F, P | $\mathcal{U}(1,5)$ | $\mathcal{U}_{\log}(10^{-12},10^{-10})$ |
| | | $y$ | Distance | V, F, P | V, F, P | $\mathcal{U}(1,3)$ | $\mathcal{U}_{\log}(10^{-2},10^0)$ |
| | | $V_e$ | Voltage | V, F, P | V, F | $\mathcal{U}(1,5)$ | $\mathcal{U}_{\log}(10^{-1},10^1)$ |
| | | $d$ | Distance | V, F, P | V, F, P | $\mathcal{U}(4,6)$ | $\mathcal{U}_{\log}(10^{-2},10^0)$ |
| B13 | $V_e = \frac{q}{4\pi\epsilon\sqrt{r^2+d^2-2dr\cos\alpha}}$ | $V_e$ | Potential | V, F, P | V, F | N/A | N/A |
| | | $\epsilon$ | permittivity | V, F, P | V, F, P | $\mathcal{U}(1,5)$ | $\mathcal{U}_{\log}(10^{-12},10^{-10})$ |
| | | $q$ | Electric charge | V, F, P | V, F | $\mathcal{U}(1,5)$ | $\mathcal{U}_{\log}(10^{-3},10^{-1})$ |
| | | $r$ | Distance | V, F, P | V, F, P | $\mathcal{U}(1,3)$ | $\mathcal{U}_{\log}(10^{-2},10^0)$ |
| | | $d$ | Distance between dipoles | V, F, P | V, F, P | $\mathcal{U}(4,6)$ | $\mathcal{U}_{\log}(10^{-2},10^0)$ |
| | | $\alpha$ | Angle | V, F, NN | V, F, NN | $\mathcal{U}(0,6)$ | $\mathcal{U}(0,\pi)$ |
| B14 | $V_e = E_f\cos\theta\left(\frac{\alpha-1}{\alpha+2}\frac{d^3}{r^2}-r\right)$ | $V_e$ | Potential (out) | V, F | V, F | N/A | N/A |
| | | $E_f$ | Magnitude of electric field | V, F, P | V, F | $\mathcal{U}(1,5)$ | $\mathcal{U}_{\log}(10^1,10^3)$ |
| | | $\theta$ | Angle | V, F, NN | V, F, NN | $\mathcal{U}(0,6)$ | $\mathcal{U}(0,\pi)$ |
| | | $r$ | Distance | V, F, P | V, F, P | $\mathcal{U}(1,5)$ | $\mathcal{U}_{\log}(10^{-2},10^0)$ |
| | | $d$ | Radius of dielectric sphere | V, F, P | V, F, P | $\mathcal{U}(1,5)$ | $\mathcal{U}_{\log}(10^{-2},10^0)$ |
| | | $\alpha$ | Polarizability | V, F, P | V, F, P | $\mathcal{U}(1,5)$ | $\mathcal{U}_{\log}(10^{-1},10^1)$ |
| B15 | $\omega_0 = \frac{\sqrt{1-\frac{v^2}{c^2}}}{1+\frac{v}{c}\cos\theta}\omega$ | $\omega_0$ | Frequency of electromagnetic waves | V, F, P | V, F | N/A | N/A |
| | | $v$ | Velocity | V, F, P | V, F, P | $\mathcal{U}(1,3)$ | $\mathcal{U}_{\log}(10^5,10^7)$ |
| | | $c$ | Speed of light | V, F, P | C, F, P | $\mathcal{U}(5,20)$ | $2.998\times10^8$ |
| | | $\omega$ | Frequency of electromagnetic waves | V, F, P | V, F, P | $\mathcal{U}(1,5)$ | $\mathcal{U}_{\log}(10^9,10^{11})$ |
| | | $\theta$ | Angle | V, F, NN | V, F, NN | $\mathcal{U}(0,6)$ | $\mathcal{U}(0,2\pi)$ |
| B16 | $E = qV_e$ $+ \sqrt{(p-qA)^2c^2+m^2c^4}$ | $E$ | Energy | V, F, P | V, F | N/A | N/A |
| | | $p$ | Momentum | V, F, P | V, F | $\mathcal{U}(1,5)$ | $\mathcal{U}_{\log}(10^{-9},10^{-7})$ |
| | | $q$ | Electric charge | V, F, P | V, F | $\mathcal{U}(1,5)$ | $\mathcal{U}_{\log}(10^{-11},10^{-9})$ |
| | | $A$ | Vector potential | V, F, P | V, F | $\mathcal{U}(1,5)$ | $\mathcal{U}_{\log}(10^1,10^3)$ |
| | | $c$ | Speed of light | V, F, P | C, F, P | $\mathcal{U}(1,5)$ | $2.998\times10^8$ |
| | | $m$ | Mass | V, F, P | V, F, P | $\mathcal{U}(1,5)$ | $\mathcal{U}_{\log}(10^{-30},10^{-28})$ |
| | | $V_e$ | Voltage | V, F, P | V, F | $\mathcal{U}(1,5)$ | $\mathcal{U}_{\log}(10^{-1},10^1)$ |
| B17 | $E = \frac{1}{2m}$ $\left(p^2+m^2\omega^2x^2\left(1+\alpha\frac{x}{y}\right)\right)$ | $E$ | Energy | V, F, P | V, F | N/A | N/A |
| | | $m$ | Mass | V, F, P | V, F, P | $\mathcal{U}(1,5)$ | $\mathcal{U}_{\log}(10^{-30},10^{-28})$ |
| | | $p$ | Momentum | V, F, P | V, F | $\mathcal{U}(1,5)$ | $\mathcal{U}_{\log}(10^{-9},10^{-7})$ |
| | | $\omega$ | Frequency of electromagnetic waves | V, F, P | V, F | $\mathcal{U}(1,5)$ | $\mathcal{U}_{\log}(10^9,10^{11})$ |
| | | $x$ | Position | V, F, P | V, F | $\mathcal{U}(1,5)$ | $\mathcal{U}_{\log}(10^{-11},10^{-9})$ |
| | | $\alpha$ | Deviation from the harmonic oscillator | V, F, P | V, F | $\mathcal{U}(1,5)$ | $\mathcal{U}_{\log}(10^{-1},10^1)$ |
| | | $y$ | Distance | V, F, P | V, F, P | $\mathcal{U}(1,5)$ | $\mathcal{U}_{\log}(10^{-11},10^{-9})$ |
| B19 | $p_f = -\frac{1}{8\pi G}$ $\left(\frac{c^4 k_f}{a_f^2}+c^2H^2(1-2\alpha)\right)$ | $p_f$ | Pressure | V, F | V, F | N/A | N/A |
| | | $G$ | Gravitational constant | V, F, P | C, F, P | $\mathcal{U}(1,5)$ | $6.674\times10^{-11}$ |
| | | $c$ | Speed of light | V, F, P | C, F, P | $\mathcal{U}(1,5)$ | $2.998\times10^8$ |
| | | $k_f$ | Variable | V, F, P | V, F | $\mathcal{U}(1,5)$ | $\mathcal{U}_{\log}(10^1,10^3)$ |
| | | $a_f$ | Distance | V, F, P | V, F, P | $\mathcal{U}(1,5)$ | $\mathcal{U}_{\log}(10^8,10^{10})$ |
| | | $H$ | Hubble's Constant | V, F, P | V, F, P | $\mathcal{U}(1,5)$ | $\mathcal{U}_{\log}(10^0,10^2)$ |
| | | $\alpha$ | Variable | V, F, P | V, F | $\mathcal{U}(1,5)$ | $\mathcal{U}(-10,10)$ |
| B20 | $A = \frac{\alpha^2 h^2}{4\pi m^2 c^2}\left(\frac{\omega_0}{\omega}\right)^2$ $\left(\frac{\omega_0}{\omega}+\frac{\omega}{\omega_0}-\sin^2\theta\right)$ | $A$ | Differential cross section | V, F | V, F, P | N/A | N/A |
| | | $\alpha$ | Fine structure constant | V, F, P | C, F, P | $\mathcal{U}(1,5)$ | $7.297\times10^{-3}$ |
| | | $h$ | Planck constant | V, F, P | C, F, P | $\mathcal{U}(1,5)$ | $6.626\times10^{-34}$ |
| | | $m$ | Electron mass | V, F, P | C, F, P | $\mathcal{U}(1,5)$ | $9.109\times10^{-31}$ |
| | | $c$ | Speed of light | V, F, P | C, F, P | $\mathcal{U}(1,5)$ | $2.998\times10^8$ |
| | | $\omega_0$ | Frequency | V, F, P | V, F, P | $\mathcal{U}(1,5)$ | $\mathcal{U}_{\log}(10^9,10^{11})$ |
| | | $\omega$ | Frequency | V, F, P | V, F, P | $\mathcal{U}(1,5)$ | $\mathcal{U}_{\log}(10^9,10^{11})$ |
| | | $\theta$ | Scattering angle | V, F, NN | V, F, NN | $\mathcal{U}(0,6)$ | $\mathcal{U}(0,2\pi)$ |

## Appendix B. License of External Code

We briefly summarize the licenses of external code we used in this study. BSD 3-Clause is used for both gplearn (Koza and Poli, 2005) (`https://gplearn.readthedocs.io/en/stable/index.html`), DSR (Petersen et al., 2020) and uDSR (Landajuela et al., 2022)(`https://github.com/brendenpetersen/deep-symbolic-optimization`). Both AFP (Schmidt and Lipson, 2011) and AFP-FE (Schmidt and Lipson, 2009) (`https://github.com/cavalab/ellyn`) use GPL ver. 2 or later. AIF (Udrescu et al., 2020) (`https://github.com/SJ001/AI-Feynman`) use MIT License, and both E2E (Kamienny et al., 2022) (`https://github.com/facebookresearch/symbolicregression`) and PySR (Cranmer, 2023)(`https://github.com/MilesCranmer/PySR`) use Apache License 2.0.

## Appendix C. Hyperparameters for Symbolic Regression Baselines

Tables 19 and 20 show the hyperparameter space for symbolic regression baselines considered in this study. The hyperparameters of gplearn (Koza and Poli, 2005) [21], AFP (Schmidt and Lipson, 2011), and AFP-FE (Schmidt and Lipson, 2009) [22] are optimized by Optuna (Akiba et al., 2019), a hyperparameter optimization framework. For E2E (Kamienny et al., 2022), we reuse the checkpoint of the pretrained model the authors provided.[23] We choose hyperparameters of other methods based on suggestions in their code and/or papers.

---

21. `https://gplearn.readthedocs.io/en/stable/reference.html#symbolic-regressor`
22. `https://github.com/cavalab/ellyn`
23. `https://dl.fbaipublicfiles.com/symbolicregression/model1.pt`

Table 19: Hyperparameter sets for symbolic regression baselines (part 1).

| Method | Hyperparameter sets |
|---|---|
| gplearn | 100 trials with random combinations of the following hyperparameter spaces:
$population\_size$: $\mathcal{U}(10^2, 10^3)$, $generations$: $\mathcal{U}(10, 10^2)$,
$stopping\_criteria$: $\mathcal{U}(10^{-10}, 10^{-2})$, $warm\_start$: {True, False},
$const\_range$: {None, $(-1.0, 1.0), (-10, 10), (-10^2, 10^2), (-10^3, 10^3), (-10^4, 10^4)$},
$max\_samples$: $\mathcal{U}(0.9, 1.0)$, $parsimony\_coefficient$: $\mathcal{U}(10^{-3}, 10^{-2})$ |
| AFP | 100 trials with random combinations of the following hyperparameter spaces:
$popsize$: $\mathcal{U}(100, 1000)$, $g$: $\mathcal{U}(250, 2500)$, $stop\_threshold$: $\mathcal{U}(10^{-10}, 10^{-2})$,
$op\_list$: {['n', 'v', '+', '-', '*', '/', 'exp', 'log', '2', '3', 'sqrt'],
['n', 'v', '+', '-', '*', '/', 'exp', 'log', '2', '3', 'sqrt', 'sin', 'cos']} |
| AFP-FE | 100 trials with random combinations of the following hyperparameter spaces:
$popsize$: $\mathcal{U}(100, 1000)$, $g$: $\mathcal{U}(250, 2500)$, $stop\_threshold$: $\mathcal{U}(10^{-10}, 10^{-2})$,
$op\_list$: {['n', 'v', '+', '-', '*', '/', 'exp', 'log', '2', '3', 'sqrt'],
['n', 'v', '+', '-', '*', '/', 'exp', 'log', '2', '3', 'sqrt', 'sin', 'cos']} |
| AIF | {$bftt$: 60, $epoch$: 300, $op$: '7ops.txt', $poly\_deg$: 3},
{$bftt$: 60, $epoch$: 300, $op$: '10ops.txt', $poly\_deg$: 3},
{$bftt$: 60, $epoch$: 300, $op$: '14ops.txt', $poly\_deg$: 3},
{$bftt$: 60, $epoch$: 300, $op$: '19ops.txt', $poly\_deg$: 3},
{$bftt$: 120, $epoch$: 300, $op$: '14ops.txt', $poly\_deg$: 4},
{$bftt$: 120, $epoch$: 300, $op$: '19ops.txt', $poly\_deg$: 4},
{$bftt$: 60, $epoch$: 500, $op$: '7ops.txt', $poly\_deg$: 3},
{$bftt$: 60, $epoch$: 500, $op$: '10ops.txt', $poly\_deg$: 3},
{$bftt$: 60, $epoch$: 500, $op$: '14ops.txt', $poly\_deg$: 3},
{$bftt$: 60, $epoch$: 500, $op$: '19ops.txt', $poly\_deg$: 3} |
| DSR | {$seed$: 1, $function\_set$: ['add', 'sub', 'mul', 'div', 'sin', 'cos', 'exp', 'log']},
{$seed$: 2, $function\_set$: ['add', 'sub', 'mul', 'div', 'sin', 'cos', 'exp', 'log']},
{$seed$: 3, $function\_set$: ['add', 'sub', 'mul', 'div', 'sin', 'cos', 'exp', 'log']},
{$seed$: 4, $function\_set$: ['add', 'sub', 'mul', 'div', 'sin', 'cos', 'exp', 'log']},
{$seed$: 5, $function\_set$: ['add', 'sub', 'mul', 'div', 'sin', 'cos', 'exp', 'log']},
{$seed$: 1, $function\_set$: ['add', 'sub', 'mul', 'div', 'sin', 'cos', 'exp', 'log', 'const']},
{$seed$: 2, $function\_set$: ['add', 'sub', 'mul', 'div', 'sin', 'cos', 'exp', 'log', 'const']},
{$seed$: 3, $function\_set$: ['add', 'sub', 'mul', 'div', 'sin', 'cos', 'exp', 'log', 'const']},
{$seed$: 4, $function\_set$: ['add', 'sub', 'mul', 'div', 'sin', 'cos', 'exp', 'log', 'const']},
{$seed$: 5, $function\_set$: ['add', 'sub', 'mul', 'div', 'sin', 'cos', 'exp', 'log', 'const']} |
| E2E | We reused the checkpoint of the pretrained model the authors provided.[23] |

Table 20: Hyperparameter sets for symbolic regression baselines (part 2).

| Method | Hyperparameter sets |
|---|---|
| uDSR | {*seed*: 1, *function_set*: ['add', 'sub', 'mul', 'div', 'sin', 'cos', 'exp', 'log', 'poly'], *batch_size*: 1000, *learning_rate*: 0.0005, *entropy_weight*: 0.03}, {*seed*: 2, *function_set*: ['add', 'sub', 'mul', 'div', 'sin', 'cos', 'exp', 'log', 'poly'], *batch_size*: 1000, *learning_rate*: 0.0005, *entropy_weight*: 0.03}, {*seed*: 3, *function_set*: ['add', 'sub', 'mul', 'div', 'sin', 'cos', 'exp', 'log', 'poly'], *batch_size*: 1000, *learning_rate*: 0.0005, *entropy_weight*: 0.03}, {*seed*: 4, *function_set*: ['add', 'sub', 'mul', 'div', 'sin', 'cos', 'exp', 'log', 'poly'], *batch_size*: 1000, *learning_rate*: 0.0005, *entropy_weight*: 0.03}, {*seed*: 5, *function_set*: ['add', 'sub', 'mul', 'div', 'sin', 'cos', 'exp', 'log', 'poly'], *batch_size*: 1000, *learning_rate*: 0.0005, *entropy_weight*: 0.03}, {*seed*: 6, *function_set*: ['add', 'sub', 'mul', 'div', 'sin', 'cos', 'exp', 'log', 'poly'], *batch_size*: 500, *learning_rate*: 0.0025, *entropy_weight*: 0.3}, {*seed*: 7, *function_set*: ['add', 'sub', 'mul', 'div', 'sin', 'cos', 'exp', 'log', 'poly'], *batch_size*: 500, *learning_rate*: 0.0025, *entropy_weight*: 0.3}, {*seed*: 8, *function_set*: ['add', 'sub', 'mul', 'div', 'sin', 'cos', 'exp', 'log', 'poly'], *batch_size*: 500, *learning_rate*: 0.0025, *entropy_weight*: 0.3}, {*seed*: 9, *function_set*: ['add', 'sub', 'mul', 'div', 'sin', 'cos', 'exp', 'log', 'poly'], *batch_size*: 500, *learning_rate*: 0.0025, *entropy_weight*: 0.3}, {*seed*: 10, *function_set*: ['add', 'sub', 'mul', 'div', 'sin', 'cos', 'exp', 'log', 'poly'], *batch_size*: 500, *learning_rate*: 0.0025, *entropy_weight*: 0.3} |
| PySR | *procs*: 5, *populations*: 10, *population_size*: 40, *ncyclesperiteration*: 500, *niterations*: 50000, *timeout_in_seconds*: 82800, *maxsize*: 50, *binary_operators*: ['*', '+', '-', '/'], *unary_operators*: ['sin', 'cos', 'exp', 'log'], *nested_constraints*: {sin: {sin: 0, cos: 0}, cos: {sin: 0, cos: 0}, exp: {exp: 0}, log: {log: 0}}, *progress*: False, *weight_randomize*: 0.1, *precision*: 32, *warm_start*: False, *turbo*: True, *update*: False |

## Appendix D. Qualitative Analysis

This section discusses qualitative analysis for the experimental results in Section 5.3, focused on the effect of introduced dummy variables on the behaviors of SR baseline methods since the SRSD-Feynman datasets with dummy variables seems extremely challenging from the solution rate and NED in Tables 4 and 5. Taking two SRSD problems (I.12.1 and II.27.16) as examples, Table 21 highlights how the randomly introduced dummy variables made changes in both the true models (equations) and the SR baseline methods' predicted equations. For the SRSD problem I.12.1 in Table 2, all the SR baselines except E2E made the perfect predictions, which completely match the true model. When introducing a random dummy variable ($x_2$ for this problem) to I.12.1, however, gplearn failed to complete the training process, and AIF, DSR, E2E, and uDSR produced little bit overcomplex symbolic expressions, including the

dummy variable ($x_2$). A similar trend can be confirmed for another SRSD problem II.27.16 in Table 9. While AFP, AFP-FE, AIF, uDSR, and PySR produced the correct symbolic expressions in terms of NED (*i.e.*, NED = 0), randomly introduced dummy variables $x_0$ and $x_1$ worsened the predictions of AFP-FE, AIF, and uDSR. Note that even though the original SRSD problem II.27.16 contains only one input variable $x_0$, two dummy variables randomly introduced as the first and second columns of the tabular dataset reindexed the original input variable $x_0$ as $x_2$ in this specific dataset due to the dummy variables.

## Appendix E. Injecting Noise to Target Variables

Following SRBench (La Cava et al., 2021), we introduce Gaussian noise with a parameter of noise level $\gamma$ to the target variables in our SRSD datasets. We inject the noise to each of the datasets separately (Eq. (5)):

$$
y_j^{\text{noise}} = f_{\text{true}}\left(X_j\right) + \epsilon, \qquad \epsilon \sim \mathcal{N}\left(0, \gamma\sqrt{\frac{1}{N}\sum_{k=1}^{N} f_{\text{true}}\left(X_k\right)}\right), \tag{5}
$$

where $1 \leq j \leq N$ and $N$ indicates the number of samples in the dataset.

Table 22 shows normalized edit distances of our baselines for noise-injected SRSD (Easy), reusing the set of noise levels in SRBench (La Cava et al., 2021) *i.e.*, $\gamma \in \{0, 10^{-3}, 10^{-2}, 10^{-1}\}$. Overall, the more the injected noise is, the more difficult it would be for the baseline models to (re-)discover the physical law in the data.

## Appendix F. Solution Rate Comparison - FSRD vs. SRSD -

Table 23 compares the solution rates of the five common baselines for the FSRD and our SRSD datasets. We can confirm that the overall solution rates for our SRSD are significantly degraded compared to those for the FSRD datasets reported in SRBench (La Cava et al., 2021) except for DSR.[24] The results indicate that our SRSD datasets are more challenging than the FSRD datasets in terms of solution rate.

## Appendix G. Limitations

### G.1 Implicit Functions

Symbolic regression generally has a limitation in inferring implicit functions, as the model infers a trivial constant function if there are no restrictions on variables. For example, $f(x,y) = 0$ is inferred as $0 = 0 \ \forall x, y$. This problem can be solved by applying the constraint that an inferred function should depend on at least two variables *e.g.*, inferring $f(x,y) = 0$ with $\frac{\partial f}{\partial x} \neq 0$ and $\frac{\partial f}{\partial y} \neq 0$, or by converting the function to an explicit form *e.g.*, $y = g(x)$. We converted some functions in the datasets into explicit forms and avoided the inverse trigonometric functions as described in Section 3.1.

---

24. Chi-squared tests showed p-values of $4.30 \times 10^{-5}$, $1.05 \times 10^{-4}$, $1.61 \times 10^{-6}$, $1.99 \times 10^{-21}$, and 0.479 for gplearn, AFP, AFP-FE, AIF, and DSR respectively.

Table 21: Examples: SRSD problems I.12.1 (top) and II.27.16 (bottom) from SRSD-Feynman (*Easy set*) to highlight how introduced dummy variables affected behaviors of the SR baselines. Coefficients are rounded for better presentation. N/A: No prediction obtained as the training process did not complete.

| Dummy var(s). ? | **No** | $x_2$ |
|---|---|---|
| True model | $x_0 \cdot x_1$ | $x_0 \cdot x_1$ |
| gplearn | $x_0 \cdot x_1$ | N/A |
| AFP | $x_0 \cdot x_1$ | $x_0 \cdot x_1$ |
| AFP-FE | $x_0 \cdot x_1$ | $x_0 \cdot x_1$ |
| AIF | $x_0 \cdot x_1$ | $0.999 \cdot x_0 \cdot x_1 + 0.159 \cdot x_2$ |
| DSR | $x_0 \cdot x_1$ | $x_0 \cdot x_1 \cdot \exp\left(-x_2^2/(x_1+x_2)\right) \cdot \cos(x_0 \cdot x_2)$ |
| E2E | $1.02 \cdot (x_0 - 1.16\text{e-}3) \cdot (x_1 - 6.88\text{e-}3)$ | $1.32\text{e+}17 \cdot (x_1 - 4.89\text{e-}3) \cdot (7.38\text{e-}18 \cdot x_0 + x_2 + 1.11\text{e-}20)$ |
| uDSR | $x_0 \cdot x_1$ | $x_1 \cdot (x_0 - \sin(x_2 \cdot \exp(x_2)/(x_0 - x_2)))$ |
| PySR | $x_0 \cdot x_1$ | $x_0 \cdot x_1$ |
| Dummy var(s). ? | **No** | $x_0$, $x_1$ |
| True model | $2.65\text{e-}3 \cdot x_0^2$ | $2.65\text{e-}3 \cdot x_2^2$ |
| gplearn | N/A | N/A |
| AFP | $2.68\text{e-}3 \cdot x_0^2$ | $2.66\text{e-}3 \cdot x_2^2$ |
| AFP-FE | $2.65\text{e-}3 \cdot x_0^2$ | $2.10\text{e-}3 \cdot x_2^2 - 0.0161$ |
| AIF | $2.65\text{e-}3 \cdot x_0^2$ | N/A |
| DSR | N/A | $4.91\text{e-}3 \cdot x_2^2 \cdot \cos(\exp(x_0))$ |
| E2E | $(0.191 \cdot x_0 - 0.0375) \cdot (\tan(0.0137 \cdot x_0 + 3.21\text{e-}3) - 7.05\text{e-}5)$ | $(2.60\text{e-}3 \cdot x_2 + 4.75\text{e-}5) \cdot (x_2 - 0.0104 \cdot \sin(1.84\text{e+}21 \cdot x_0 - 2.40\text{e-}11 \cdot x_1 - 9.12\text{e+}26 \cdot x_3 + 1.36) + 0.0398)$ |
| uDSR | $2.65\text{e-}3 \cdot x_0^2$ | $2.65\text{e-}3 \cdot x_1 \cdot x_2^2/(x_1 - x_2)$ |
| PySR | $2.65\text{e-}3 \cdot x_0^2$ | $2.65\text{e-}3 \cdot x_0^2$ |

## G.2 Noise Injection

When applying machine learning to real-world problems, it is often true that the observed values contain some noise. While we follow La Cava et al. (2021) and show experimental results for our SRSD datasets with noise-injected target variables in Appendix E, these

Table 22: Normalized edit distances of baselines for noise-injected SRSD (Easy) datasets with different noise levels.

| Noise Level ($\gamma$) \ Method | gplearn | AFP | AFP-FE | AIF | DSR |
|:---:|:---:|:---:|:---:|:---:|:---:|
| 0 | 0.876 | 0.703 | 0.712 | 0.646 | **0.551** |
| $10^{-3}$ | 0.928 | 0.799 | 0.814 | **0.797** | 0.820 |
| $10^{-2}$ | 0.940 | 0.824 | 0.880 | 0.870 | **0.793** |
| $10^{-1}$ | 0.948 | **0.823** | 0.960 | 0.882 | 0.841 |

Table 23: Solution rates of the common baselines between FSRD and SRSD-Feynman datasets.

| Dataset \ Method | gplearn | AFP | AFP-FE | AIF | DSR |
|:---:|:---:|:---:|:---:|:---:|:---:|
| FSRD (Udrescu and Tegmark, 2020) | 15.7% | 20.41% | 26.08% | **53.0%** | 19.1% |
| SRSD (Ours) | 1.67% | 5.83% | 6.67% | 9.17% | **15.8%** |

aspects are not thoroughly discussed in this study, such discussions can be a separate paper built on this work and further engage studies of symbolic regression for scientific discovery.

### G.3 Dependency on sympy

Similar to SRBench (La Cava et al., 2021), the implementation of our evaluation pipeline has a significant dependency on sympy. Specifically, when computing edit distance between the predicted and true expressions and solution rate, our evaluation pipeline builds equation trees based on the tree structure of the expressions used in sympy after converting the expressions to floating-point approximations. Our use of edit distance and solution rate is based on our observation and an assumption that sympy consistently maps a given equation to the unique equation tree, handling algebraic properties so that we can compute edit distance between the true and estimated equation trees consistently. We also acknowledge that sympy may fail to process too complex expressions, and some symbolic regression methods may produce such solutions. However, since the interpretability of the prediction is a key property of symbolic regression, such overcomplex expressions should not be desired by non-ML users and will result in NED = 1 for SRSD problems considered in this study.

