# OpenReview forum: "Rethinking Symbolic Regression Datasets and Benchmarks for Scientific Discovery"
_DMLR — Accepted by DMLR_

### Review · Reviewer_KcAx · 2023-11-27

**Recommendation:** 3
**Confidence:** 3

**Summary Of Contributions:**

The paper presents a dataset for the task of symbolic regression of scientific discoveries, which is the task of discovering formulas from tables of experiment results that demonstrate a certain scientific law (specifically, in this paper they focused on physics laws).
The authors identified issues with existing, well-known, datasets that were used for this task, such as equivalency in formulae that causes multiplicities, or the issue of mistreating scientific constants as variables. Then, they suggested solutions to these problems and implemented them in the new suggested dataset. The authors also created a new challenge by sampling dummy variables to test if the models designed for this task are able to distinguish those from true variables.
In addition, the authors propose a new metric to evaluate models for the task that not only measures if the predicted formula is able to predict the correct values, but also measures the distance between the true formula and the predicted one. They also add a user study that demonstrates their evaluation metric is more aligned with human annotations that judge how close the predicted formula is to the real formula.
To conclude, the authors compare between baseline and SOTA algorithms to solve the task on the new dataset and present their conclusions from the empirical investigation.

**Strengths:**

The paper presents solid research that was made to improve an existing benchmark for the task of symbolic regression of scientific discoveries.
1. The authors identified significant issues with the current datasets, explained them clearly, and demonstrated with examples why they are serious limitations for evaluating the models designed for this task.
2. The authors design changes and implement them on the existing dataset to fix the limitations stated above, and add another dataset that presents another challenge to models, which will increase the robustness of the models designed for this task.
3. The paper presents enough empirical evidence that their suggested dataset is challenging and that the new suggested evaluation metric is useful for the task.

**Broader Impact Concerns:**

No concerns at all.

**Claims And Evidence:**

Except for the missing statistical test results, the empirical investigation made in this paper is thorough and sufficient.

**Datasets And Benchmarks:**

The dataset link is included in the paper, together with sufficient details of usage and data collection. There are no special concerns about ethical or responsible use. The paper describes in great detail all the necessary processes of data collection and annotation.

**Extended Submissions:**

This is not an extended version as far as I can tell.

**Limitations:**

1. The strong negative correlation between human judgments and NED indicates an inverse relationship, necessitating the inversion of the NED score. However, this appears counterintuitive given its inherent meaning. How do the authors rationalize this outcome?
2. The strong correlation observed between NED scores and human judgment, surpassing that with R^2, is unsurprising. This high correlation is attributed to humans evaluating formula similarity, a criterion not captured by R^2, as clarified by the authors when introducing their novel evaluation metric.
3. While significance claims regarding model performance and differences were asserted across various comparisons, statistical tests were exclusively reported for the user study (i.e., p-value). This raises concerns about the comprehensive validation of these claims in the absence of statistical scrutiny for the other comparisons.
4. Another limitation is that the scope of the dataset and the annotation principles described in the paper only fit formulas for natural and physics laws, it does not consider probabilistic formulae for example. I am not sure this is a true limitation since the authors describe clearly this is not within the scope of this paper, but I would recommend adding a section about further possible evolutions for this dataset according to the annotation principles of the paper.

**Requested Changes:**

Statistical results (i.e., p-value) should be added to the paper for each claim of significance made in the paper:
"uDSR significantly outperforms all the other baselines"
"PySR produced more solutions structurally close to the true models than the other baseline methods and significantly improved the other
baseline methods in terms of solution rate and NED."
"We can confirm that the overall solution rates for our SRSD are significantly degraded compared to those for the FSRD datasets reported in SRBench (La Cava et al., 2021)."

---

### Review · Reviewer_WPvE · 2023-12-04

**Recommendation:** 3
**Confidence:** 1

**Summary Of Contributions:**

The paper introduces significant advancements in the field of Symbolic Regression (SR) for scientific discovery. The authors have meticulously identified and addressed the limitations in existing SR datasets and benchmarks, leading to the creation of new datasets (SRSD) that are more reflective of real-world physics experiments. Additionally, the introduction of a new evaluation metric, the normalized edit distance (NED), offers a novel way to assess the performance of SR methods in terms of their structural accuracy. These contributions collectively enhance the practical applicability and evaluation rigor of SR in scientific research.

**Strengths:**

1. Significance of Contribution: The paper addresses a critical gap in SR research by providing more realistic and applicable datasets, along with a new evaluation metric.

2. Relevance to Broader Community: The proposed datasets and metrics have broad applicability across various scientific disciplines that utilize SR for data modeling and discovery.

3. Quality of Research: The research methodology is sound, with a clear rationale behind each dataset modification and metric introduction.

4. Clarity of Paper: The paper is well-structured and articulates complex concepts in an accessible manner.

**Broader Impact Concerns:**

The paper sufficiently addresses the ethical implications of its work, particularly in the responsible use of datasets for scientific discovery. However, a more detailed discussion on the potential misuse of SR, especially in sensitive scientific areas, and guidelines for ethical use would strengthen this aspect.

**Claims And Evidence:**

1. Realism and Applicability of SRSD Datasets: The paper claims that the newly proposed SRSD datasets are more realistic and applicable for scientific discovery compared to existing datasets. This claim is substantiated with detailed analyses, such as the identification of limitations in existing datasets (like the oversimplification of sampling processes and lack of physical meaning in symbolic expressions). The authors provide clear evidence through the meticulous recreation of 120 datasets based on Feynman's physics lectures, ensuring more realistic sampling ranges and variable treatments. However, the evidence primarily supports the claim within the context of physics. Extending this evidence to include datasets from other scientific fields would strengthen the claim's generalizability.

2. Effectiveness of NED Metric: The introduction of the normalized edit distance (NED) as a new evaluation metric is a central claim. The authors present this metric as a more effective tool for assessing the performance of SR methods, particularly in capturing structural accuracy. The evidence provided includes benchmark experiments showing that NED correlates significantly more with human judgment compared to existing SR metrics.

3. Superiority of Certain SR Methods on SRSD Datasets: The authors claim that specific SR methods, particularly uDSR and PySR, performed best on the SRSD-Feynman datasets. This is evidenced by experimental results showing these methods' superiority in terms of R2-based accuracy and solution rates. A deeper dive into why these methods outperform others on the SRSD datasets could offer valuable insights into the characteristics of effective SR methods for scientific discovery.

4. Impact of Dummy Variables on SR Methods: The paper asserts that the presence of dummy variables significantly challenges the SR methods tested. The evidence for this claim is shown in the performance differences on datasets with and without dummy variables. However, while the results indicate a general trend of decreased performance, a more detailed examination of how different methods handle dummy variables—potentially including an analysis of their internal decision-making processes—would provide stronger evidence for this claim.

**Datasets And Benchmarks:**

The paper provides detailed information on data collection, organization, and the creation of the new SRSD datasets. There is clear documentation on the intended use of these datasets.  The code https://github.com/omron-sinicx/srsd-benchmark/tree/main is available.

**Extended Submissions:**

This section is not applicable as the paper does not appear to be an extended version of a previously published work. It seems to present original research and findings.

**Limitations:**

1. The paper's title is "Rethinking Symbolic Regression Datasets and Benchmarks for Scientific Discovery" while this paper only focuses on Physics. The paper could benefit from more diverse application examples beyond physics to demonstrate the broader applicability of the proposed datasets and metrics.

2. Experimental Scope: The current experiments focus on evaluating the new datasets using a select group of established symbolic regression (SR) methods. To enhance the robustness of the datasets' validation, it would be beneficial to expand these experiments to incorporate a more diverse array of SR techniques, particularly those embodying different methodologies or more recent advancements. While the paper's benchmarking approach effectively utilizes several well-regarded baselines (such as gplearn, AFP, AFP-FE, AI Feynman (AIF), DSR, a Transformer-based method (E2E), uDSR, and PySR), it would be advantageous for the authors to include a broader spectrum of common baseline methods found in GitHub repositories. This inclusion would facilitate ease of use and adoption by future researchers in the field.

3. Computational Resource Analysis: The paper mentions the use of high-performance computing (HPC) resources for experiments but does not delve into the specifics of computational efficiency or resource demands. Given the diverse resource capabilities of potential users, an analysis of the computational requirements, including runtime and memory usage for different SR methods on the proposed datasets, would be valuable. This aspect is particularly crucial for understanding the practical applicability of the SRSD datasets in various research settings.

4. Handling of Dummy Variables: The authors assert the challenges dummy variables pose to SR methods, but the analysis mainly highlights the failure of baseline methods to filter out these variables. A more detailed exploration of why these methods struggle with dummy variables, possibly including a breakdown of how each method reacts to the presence of dummy variables, would provide deeper insights. This could also guide future improvements in SR methods to handle such scenarios more effectively.

5. I am also curious about the performance of GPT4 on the new proposed benchmark.

**Requested Changes:**

Broaden the Scope of Datasets:
Change: Include datasets from a wider range of scientific disciplines beyond physics to demonstrate the SRSD datasets' applicability across various fields.
Criticality: Strengthening. While the current focus on physics is robust, broadening the scope would significantly enhance the paper's relevance and applicability to a wider research community.

Detailed Computational Resource Analysis:
Change: Include a thorough analysis of the computational demands for using the SRSD datasets and NED metric, such as runtime and memory requirements.
Criticality: Strengthening. This information is important for readers to assess the practicality of implementing the proposed methods in various settings.

Comprehensive Analysis of Dummy Variable Handling:
Change: Present a more detailed examination of how different SR methods respond to dummy variables, possibly including a breakdown of their decision-making processes.
Criticality: Critical. This would substantiate the claim regarding the challenge of dummy variables and provide insights into improving SR methods.

Detailed Computational Resource Analysis:
Change: Include a thorough analysis of the computational demands for using the SRSD datasets and NED metric, such as runtime and memory requirements.
Criticality: Strengthening. This information is important for readers to assess the practicality of implementing the proposed methods in various settings.

Evaluate the Performance of GPT-4 on the New Benchmark:
Change: Test and report the performance of GPT-4, a state-of-the-art language model, on the newly proposed SRSD benchmark.
Criticality: Strengthening. This would add a contemporary perspective to the benchmark evaluation, showcasing its applicability to cutting-edge AI models.

---

### Review · Reviewer_oE2q · 2023-12-14

**Recommendation:** 3
**Confidence:** 2

**Summary Of Contributions:**

This paper proposes a new benchmark for the symbolic regression for scientific discovery (SRSD) task -- the task of fitting a mathematical expression to a given dataset for scientific discovery. Note that recently La Cava et al. (2021) constructed a benchmark SRBench combining multiple datasets and comparing several SR methods. This work claims that (1) SRBench is oversimplified due to values being sampled from limited domains and that variables and constants in these datasers have no physical meanings, and (2) there is lack of suitable metrics to evaluate SRSD methods, as the current metrics are too coarse-grained. To address these, this work proposes new datasets (easy, medium, and hard based on the complexity of the problem) with updated sampling ranges that consider physical phenomenon, and introduces an evaluation metric based on normalized edit distance on the tree structure. They also introduce SRSD datasets with dummy varibles to test the robustness of SR methods. I am leaning to accept this work.

**Strengths:**

(1) Presentation is extremely clear, and the paper is well-organized.

(2) Introducing a dataset with dummy variables to measure robustness of SR methods seems novel with respect to existing works and broadly helpful for the scientific community.

(3) It is valuable that they introduced a metric that goes beyond binary accuracy and measure how structurally close ground truth and predicted equations are. They also conduct a user study to measure how aligned the metric is with human judges.

**Broader Impact Concerns:**

I do not have any broader impact concerns about the work.

**Claims And Evidence:**

Claims are sound and necessary evidence is provided to the most part.

**Datasets And Benchmarks:**

(1) Sufficient detail on data collection and organization: Included in Section 3.
(2) Availability and maintenance: Yes
(3) Ethical and responsible use: Included in the "Broader Impact Statement".
(4) Documentation and intended uses: Yes
(5) URL for reviewer access to the dataset: Included in github link
(6) Hosting, licensing and maintenance plan: Yes
(7) Sufficient detail to support reproducibility: Work is mostly producible except the references on how they determined the sampling ranges so that they fit the physical phenomenon. If authors could provide details on how this process is conducted in Appendix, that would be helpful.

**Extended Submissions:**

This is not presented as an extended submission.

**Limitations:**

(1) Although they describe the issues with existing datasets in Section 3.1, it is not clear which datasets suffer from which of these issues. Hence, it is hard to determine the contribution over the existing work -- a figure would be helpful here (see Table 1 in https://arxiv.org/pdf/2203.06823.pdf as an example where columns would be "issues" instead of "Tasks").

(2) Work is mostly producible except the references on how they determined the sampling ranges so that they fit the physical phenomenon. If authors could provide details on how this process is conducted in Appendix, that would be helpful.

**Requested Changes:**

Please address the limitations I point out in the Weaknesses section.